# Sex Differences in Neuropathy: The Paradigmatic Case of MetFormin

**DOI:** 10.3390/ijms232314503

**Published:** 2022-11-22

**Authors:** Federica De Angelis, Valentina Vacca, Jessica Tofanicchio, Georgios Strimpakos, Giacomo Giacovazzo, Flaminia Pavone, Roberto Coccurello, Sara Marinelli

**Affiliations:** 1National Research Council (CNR), Institute of Biochemistry and Cell Biology, 00015 Monterotondo, RM, Italy; 2European Center for Brain Research—IRCCS Santa Lucia Foundation, 00143 Rome, RM, Italy; 3Neurobiology, Department of Biology and Biotechnology “Charles Darwin”, Sapienza University of Rome, 00185 Rome, RM, Italy; 4National Research Council (CNR), Institute for Complex System (ISC), 00185 Rome, RM, Italy

**Keywords:** neuropathic pain, allodynia, metformin, sex differences, macrophages, autophagy, AMPK, TNFα, leptin, neurofilaments

## Abstract

As a widely prescribed anti-diabetic drug, metformin has been receiving novel attention for its analgesic potential. In the study of the complex etiology of neuropathic pain (NeP), male and female individuals exhibit quite different responses characterized by higher pain sensitivity and greater NeP incidence in women. This “gender gap” in our knowledge of sex differences in pain processing strongly limits the sex-oriented treatment of patients suffering from NeP. Besides, the current investigation of the analgesic potential of metformin has not addressed the “gender gap” problem. Hence, this study focuses on metformin and sex-dependent analgesia in a murine model of NeP induced by chronic constriction injury of the sciatic nerve. We investigated sexual dimorphism in signaling pathways involved by 7 days of metformin administration, such as changes in AMP-activated protein kinase and the positive regulation of autophagy machinery, discovering that metformin affected in a sexually dimorphic manner the immunological and inflammatory response to nerve lesion. These effects were complemented by morphological and adaptive changes occurring after peripheral nerve injury. Altogether these data can contribute to explaining a number of potential mechanisms responsible for the complete recovery from NeP found in male mice, as opposed to the failure of long-lasting recovery in female animals.

## 1. Introduction

The use of the glucose-lowering metformin (MTF) as first-line medication in type 2 diabetes (T2D) is widely acknowledged [1,2], as well as its clinical use as an agent to limit the impact of antipsychotic-induced body weight gain and dyslipidemia [3]. In parallel to T2D therapeutic management, the administration of MTF is associated with reduced cardiovascular risk [4] and MTF is also prescribed for the treatment of gestational diabetes mellitus and polycystic ovary syndrome [5]. Moreover, because of the beneficial effects reported on osteoarthritis, cancer and diabetes-associated dementia [6,7,8,9], MTF is considered to possess anti-aging properties [10] mainly exerted throughout autophagy stimulation [11]. Basically, MTF can inhibit mitochondrial complex I, thus decreasing cellular respiration and ATP levels and stimulating AMP-activated protein kinase (AMPK) in the liver (i.e., hepatocytes) [12,13]. In turn, AMPK activation is known to depress the activity of the mechanistic target of rapamycin complex 1 (mTORC1) via multiple steps [14,15]. Previous papers demonstrated that neuropathic pain (NeP) could be counteracted and its chronicization prevented by rapamycin administration, which is able to inhibit targeting mTORC1 signaling and promote autophagy [16,17,18,19]. Together, AMPK activation, negative regulation of mTORC1 activity and decrease of activation of extracellular regulated protein kinase (ERK) pathways in sensory neurons were mechanisms identified in a couple of seminal studies in which MTF administration was shown to abolish nerve injury or spinal nerve ligation-induced allodynia [20,21]. Interestingly, different exogenous factors that can stimulate autophagy and AMPK activators, such as physical exercise [22] and caloric restriction (CR) [23], were shown to provide relief from peripheral nerve injury-induced tactile allodynia and neuropathy.

However, while CR is a well-known autophagy inducer/enhancer [24], mTORC1 is a well-recognized negative regulator of the autophagy machinery [25]. Thus, the enhancement of autophagy can be viewed as a key mechanism by which MTF plays its pharmacological action and achieve therapeutic efficacy [26].

The idea of defective autophagy in both the genesis and exacerbation of NeP has received repeated experimental confirmation. Disrupted autophagy has been described in a mouse model of spinal nerve ligation-induced NeP [27], as well as acutely enhanced autophagy after spinal nerve injury as a defensive mechanism triggered in dorsal root ganglion (DRG), which suggests the use of autophagy inducers (e.g., rapamycin) as a therapeutic option to prevent the exacerbation of NeP [28]. We further confirmed this notion by demonstrating that allodynia is exacerbated in a transgenic mouse model of defective autophagy (i.e., Beclin-1-regulated autophagy (Ambra1 (+/gt) [19], and that CR is able to promote Schwann cell autophagy and remyelination of peripheral nerve after chronic constriction injury (CCI) via AMPK pathway activation in Ambra1 (+/gt) mice [23]. In parallel, the increasing epidemiological evidence for a major incidence of pain sensitivity and chronic NeP in women [29] has been confirmed in the last years by several preclinical models. Multiple mechanisms underlying sex differences in pain experience, lower tolerance to pain in females and sexually dimorphic efficacy of drug intervention have been either hypothesized or demonstrated. For instance, sex affects the response of the immune system, as shown by the role of the adaptive immune system and higher infiltration of cluster of differentiation 4 (CD4)+ T-lymphocytes after peripheral nerve injury [30], or regarding the contribution of regulatory T-cells (Tregs)-mediated suppression of microglia activation as well as the alteration of macrophage activities and pain hypersensitivity in female mice [31]. We also investigated sex differences in NeP, providing evidence that microglia homeostasis follows in female mice a different temporal pattern of activation after peripheral nerve lesion (i.e., chronic constriction injury, CCI), with a late and persistent glial cells activation and incomplete functional recovery [32,33]. Next, we showed that T cells are infiltrated to a greater extent in the peripheral nerve of female mice that underwent CCI [34].

Against this background and our previous reports on the role of autophagy and sex differences in pain processing, we tested the hypothesis that, as an autophagy inducer, MTF may improve Schwann cells-autophagy affecting the early phase of Wallerian degeneration (WD) in mice with neuropathy, providing analgesia and neuroprotection after CCI-induced allodynia. In particular, we investigated the temporal evolution of allodynia after subchronic (7 days) MTF administration in male and female mice and the possible sexual dimorphism in signaling pathways involved in both MTF mechanism of action (i.e., AMPK) and positive molecular regulation of autophagy machinery (i.e., the mammalian silent information regulator 2 homolog, SIRT1). The present study also investigated morphological and adaptive changes occurring after CCI of the sciatic nerve and peripheral nerve lesion by comparing the impact of MTF therapy and different expression of large myelinated A-β fiber neurons (e.g., NF200), peripheral myelin protein 22 (PMP22), myelin protein zero (P0), Schwann cells proliferation (Cdc2) and macrophages (CD11b). Finally, changes in inflammatory markers (cytokines and chemokines) with specific analysis of TNFα levels have also been analyzed.

## 2. Results

### 2.1. Metformin Treatment, NeP Management, and Sex-Differences

The interval of the first seven days after peripheral nerve lesion is of critical importance not only for the autophagy of Schwann cells (SCs) and regulation of Wallerian degeneration (WD) but also for nerve regeneration [35] and pain response [19]. To understand whether chronic pain and nerve recovery in male and female mice would benefit from the potentiation of autophagy, we used MTF as an autophagy inducer [26]. We administered either MTF (200 mg/kg intraperitoneally (IP)) or vehicle (NaCl) each day for 7 consecutive days in male and female CD1 mice. Figure 1A shows a robust enhancement of paw withdrawal threshold and improvement of allodynia response in MTF-administered male and female mice from day 3 (D3) to D12 with respect to vehicle-administered mice, thus showing long-lasting MTF-induced analgesia (all statistics are reported in the legend).

However, sex differences emerged as the time course of functional recovery from neuropathy that was shown before by male mice (D60), while allodynia was not different in female mice regardless of the MTF treatment. In consideration of the metabolic effects that MTF can produce on body weight, energy balance and glucose disposal [36], we monitored body weight (BW), body temperature, glycemia, triglycerides and vitamin B12 levels (Figure 1B–G). As for the dose selected, as well as for the whole duration of MTF administration, no relevant alterations in male and female mice were observed when compared with vehicle-administered animals. On the other hand, CCI produced changes in glucose and triglyceride levels that confirmed our previous data [23]. Of interest, peripheral nerve injury produced significant changes in B12 levels, which were decreased after CCI regardless of sex and MTF treatment (Figure 1F–G).

### 2.2. Effects on Axonal and Myelin Degeneration after Peripheral Nerve Lesion

By immunofluorescence confocal analysis, it was ascertained whether MTF might affect WD-induced myelin and axonal changes in a sex-dependent manner. Hence, we examined sciatic nerves from CCI vehicle- and MTF-administered male and female mice collected 7 days after nerve lesion (Figure 2, naïve mice; Appendix A). Double staining for NF200 (i.e., intermediate neurofilament constituting an axonal structural protein of the cytoskeleton and a marker of myelinated nerve fibers) and P0 (i.e., myelin protein zero, the most abundant myelin protein; Figure 2A), disclosed important differences in NF200 staining between CCI vehicle- and CCI MTF-treated male mice. Indeed, while the former neurofilaments appear disrupted, damaged and aggregated, in CCI MTF-treated male mice, neurofilaments are intact and uninjured, as also confirmed by RGB analysis (i.e., Red Green Blue analysis [37] (all statistics are reported in legends to figure). No substantial differences were observed for the myelin protein P0 between vehicle- and MTF-treated male and female mice, and several aggregates were detectable in both sexes, although the presence of myelin ovoids in MTF-treated is also indicative of advanced WD [38]. RGB analysis (all statistics are reported in legend) confirmed the lack of differences in P0 expression between MTF-treated male and female mice. By contrast, significant differences were found in comparison with naïve mice, as previously confirmed [19]. Figure 2C shows the expression (i.e., by double staining) of S100b (an SC marker) and pmp22 (peripheral myelin protein 22), in which emerges differences in myelin degeneration regardless of MTF treatment and sex, as further confirmed by the RGB analysis (Figure 2E). The identification of positive cells for Cdc2 (Cyclin-dependent kinase 2, a marker of proliferation-Figure 2D) showed a significant decrease of proliferative cells in male and female MTF-treated mice. Since MTF is an autophagy enhancer exerting a key role in cell survival and control of both cell cycle progression and arrest [39], an inhibitory mechanism underlying MTF treatment may be hypothesized. On the other hand, by cell counting (total nuclei in the image; Figure 2E), we observed a significant enhancement in MTF-treated male and female mice, suggesting a higher incidence of immune cells, as thereafter investigated.

### 2.3. Effects on the Regulation of Cell Energy Status

AMPK and silent information regulator T1 (SIRT1) mutually interact and regulate each other, establishing a functional partnership [40] and sharing many common molecules and processes, such as mTOR, as part of the autophagy machinery [41]. Since MTF affects both AMPK and SIRT1, we evaluated (by WB and ELISA analysis on sciatic nerves lysates) the time-course of AMPK and SIRT1 activation after nerve lesion and the impact of MTF administration in male and female mice (Figure 3). As previously shown [23], 7 days after peripheral nerve lesion (i.e., CCI), there is the highest AMPK activation (phospho-pAMPK) in vehicle-treated male mice (Figure 3A). Moreover, as also previously detected in animals that underwent caloric restriction [23], MTF administration produced an early and significant enhancement of pAMPK (day 3), thus suggesting the induction of premature autophagy. Vehicle-treated female mice showed the highest pAMPK levels 3 days after CCI. By contrast, MTF-treated female mice showed a significant decrease of pAMPK in all time points evaluated, as compared to vehicle-treated female mice. SIRT1 time-course after nerve lesion was found to immediately decrease after injury (6 h) in vehicle-treated male mice, as compared to SIRT1 levels showed by non-CCI naïve mice. Later, SIRT1 levels gradually increased, reaching a plateau around D7 after CCI, while MTF administration prevented the initial decrease facilitating the progressive increase of SIRT1 during the following time course. Vehicle-treated female mice (Figure 3B) showed SIRT1 levels similar to MTF-treated male mice. No differences were found 6 h after lesion as compared to non-CCI animals, while a gradual and significant enhancement was detected during the following time points. MTF-treated female mice showed a slight change in SIRT1 levels in comparison to vehicle-treated mice.

### 2.4. Regulation of Inflammation

Peripheral nerve degeneration following injury is a process characterized by different stages and inflammatory biomarkers, in particular by cytokines and chemokines associated with the activation of the immune response. To assess the impact of MTF administration on peripheral nerve injury-induced inflammation, lysates of sciatic nerves were processed (i.e., by a microchip antibody array) to acquire information about 40 inflammatory mediators potentially involved. Table 1 reports a list of significantly changed inflammatory agents, as compared to non-CCI naïve mice. Seven days after CCI, the following agents were found upregulated in vehicle-treated mice: Eotaxin, IL-1b, IL-2, IL-3, IL-6, MCP-1, MIP-1g, TCA-3, TIMP-1, sTNF RI, and sTNF RII. On the contrary, in the same group of vehicle-treated mice, the following agents were found to be downregulated: Eotaxin-2, IL-10, IL-17, I-TAC, TECK, and TNFa. MTF administration produced in male mice the upregulation of the agents Eotaxin, IL-b, IL-2, IL-3, IL-6, LIX, Lymphotactin, MCP-1, MIG, MIP-1g, TECK, TIMP-1, and sTNF RI, while downregulated the following: Eotaxin-2, GCSF, IL-10, IL-17, I-TAC, and sTNF RII. Thus, in injured CCI male mice, MTF administration produced changes similar to those observed in vehicle-treated CCI male mice except for the overall decrease in pro-inflammatory interleukins such as IL-1b, IL-2, IL-3, and IL-6, together with the increase in chemokines level such as LIX, Lymphotactin, MCP-1, MIG, TECK, and TNFa, suggesting a general regression of the inflammatory status associated to an increase of chemotactic action. The following mediators were found upregulated in vehicle-treated CCI female mice: CD30L, Eotaxin, Eotaxin-2, Fas Ligand, GCSF, GM-CSF, IL-3, IL-4, IL-6, IL-9, IL-10, IL-12p40p70, IL-13, Leptin, LIX, MCSF, MIG, MIP-1g, RANTES, TCA-3, TECK, TIMP-1, TIMP-2, sTNF RI, and sTNF RII. Moreover, in the same experimental group, BLC was the only mediator to be found to be downregulated. MTF administration in CCI female mice upregulated the following mediators: CD30L, Eotaxin, Eotaxin-2, Fas Ligand, GCSF, GM-CSF, GM-CSF, IL-1a, IL-1b, IL-2, IL-3, IL-4, IL-6, IL-9, IL-10, IL-12p40p70, IL-13, Leptin, LIX, MCSF, MIG, MIP-1g, RANTES, TCA-3, TECK, TIMP-1, TIMP-2, and sTNF RI. As for vehicle-treated CCI female mice, also for MTF-treated CCI female mice, we observed only BLC to be downregulated. By comparing the effects of MTF treatment with its control group in CCI female mice, we observed a general increase of interleukins and chemokines, indicative of higher immune system activation.

### 2.5. Macrophages Activation after CCI

The activity of resident macrophages, together with SC autophagy, are upstream events required for triggering WD [42]. We found an increase in chemokines release and chemotaxis of circulating macrophages during the first days after the lesion and a higher response to MTF treatment (Table 1). Hence, we investigated the impact of MTF administration on macrophages, and the IF analysis of CD11b (a macrophage marker) positive cells (Figure 4A,B) revealed a significant expression of macrophages in CCI male mice, particularly in female mice. To further corroborate the effect of macrophage stimulation induced by MTF treatment, we assessed the time course of TNFα expression on sciatic nerve lysates (Figure 4C). We considered the following time points: baseline (tissue derived from naïve animals), 6 h, 24 h, D3, and D7. These time points of analysis allow us to evaluate the fluctuation of TNFα that has been taken into consideration as a response to neuropathy, treatment and sex, in combination or alone. TNFα has been reported to change very quickly after inflammation [43], and since the treatment started immediately after ligature, we could recognize whether MTF interferes with this pro-inflammatory marker.

We found a slight decrease in TNFα expression in vehicle-treated CCI male mice; the TNFα expression in all time points was considered as compared to non-CCI mice. By contrast, we found TNFα expression significantly increased in MTF-treated CCI male mice, from D3 after CCI up to D7. Vehicle-treated CCI female mice showed incremental levels of TNFα expression starting 24 h after CCI up to the end of observation. By contrast, in MTF-treated CCI female mice, such an increase was maximal at D7 after nerve injury. Moreover, we used a web-based bioinformatics tool named Ingenuity Pathways Analysis (IPA) to link information about genes, proteins, chemicals and drugs and gain a deeper understanding of the relationship between the data collected. An interactive network portraying data against a biological context is shown (Figure 4E). Two pathways depict the response to nerve injury (saline vs. naïve; grey-grey pathway) and the effects of MTF treatment on molecules involved in nerve injury (MTF vs. SAL, pink-grey pathway). In red are portrayed molecules found upregulated, while in green are those which were downregulated. The IPA analysis validated the hypothesis that, by acting on AMPK and SIRT1, MTF administration can affect different pro-inflammatory agents.

## 3. Discussion

This study aimed to investigate the anti-allodynic potential provided by MTF administration in male and female animals, thus assessing the “gender gap” in pain processing and the sex-dependent responses to MTF-induced analgesia in NeP [44]. After seven days of subchronic MTF administration, a similar analgesic response to CCI-induced allodynia was observed in male and female mice. Nevertheless, by following over time the evolution of the analgesic response, it emerged that MTF treatment provided in female mice only temporary relief. Indeed, the response to mechanical allodynia and the changes in nociception threshold were followed for 60 days after peripheral nerve injury, showing that the analgesia provided in female mice was not only transitory but abolished from D30 onward, becoming identical to the response exhibited by vehicle-treated female mice. By contrast, the anti-allodynic response in MTF-treated male mice persisted up to D30 and significantly increased from D50 up to D60, finally matching the same withdrawal threshold of the contralateral paw showed by vehicle-treated male and female mice. In other words, an incomplete and temporary recovery (i.e., female mice) against a complete functional recovery and long-lasting lack of allodynia response (i.e., male mice) were observed.

Hence, these data disclose a striking difference in terms of analgesic potential and attenuation of NeP after MTF treatment in male and female mice. How could we account for these sex-dependent differences produced by MTF-based therapy?

The first important alteration found was morphological. As components of the intermediate filaments in neurons, neurofilaments (NFs) are part of the axonal cytoskeleton and contribute to maintaining the structural integrity of axons as well as neurons’ shape and function of axonal transport [45,46]. Among the different types of neurons composing the dorsal root ganglia (DRGs), there are both large NF200-positive neurons for proprioception and small NF200-positive neurons transmitting pain [47,48]. NF200 is a highly reliable biomarker of neuronal damage in a variety of neurological and neurodegenerative diseases (e.g., Charcot-Marie-Tooth disease, giant axonal neuropathy, Alzheimer’s disease and frontotemporal dementia) and in spinal cord injury [48,49]. While peripheral nerve injury (i.e., CCI) disrupted NFs morphological integrity in vehicle-treated male mice, MTF treatment was able to stimulate axon regeneration and, potentially, remyelination of peripherally damaged nerves.

During the first days from the start of WD, myelin debris and degenerated axons are found to be aggregated in Schwann cells. This accumulation, if prolonged, facilitates inflammation, delaying the recovery process as well as promoting chronic pain [19]. In this context, the rapid removal of aggregates and myelin clearance is highly desirable. The first line of cells contributing to this stage are Schwann cells and macrophages, which both can be affected by autophagy modulation [19,50]. Confocal analysis of myelin morphology from nerve tissues validated that, 7 days after injury, in both males and females, there are myelin aggregates (P0 and PMP22), while depletion of PMP22 was identified only in female mice, thus suggesting a specific susceptibility of this protein in the female sex after injury of the peripheral nervous system. By virtue of its pro-autophagic activity, we expected, after MTF administration, a robust decrease in myelin aggregates, which instead were apparently unmodified. Of note, although cdc2-positive cells (proliferating cells) decreased after MTF treatment in both sexes, the total cell count (nuclei) was strongly enhanced.

Within this context, and considering the mechanisms of action so far identified to describe the effects of MTF administration [12,13], we assessed MTF-induced AMPK activation (i.e., phosphorylation and, therefore, autophagy promotion) in male and female mice at 6 h, 24 h, day 3 and day 7 after CCI. It should be observed that AMPK is an “energy sensing” serine/threonine enzyme that is triggered by the increase in the intracellular AMP:ATP ratio, acting as a master regulator of cellular energy status, mitochondrial metabolism and energy homeostasis [51].

The clinical interest in MTF-based therapy has been recently expanded to include several neurological disorders, mainly because of the parallel between some neuropathological mechanisms recognized in both types of diabetes (type 1 and type 2) and neurodegenerative diseases such as Alzheimer’s disease [52]. Indeed, energy dysfunction, failure in AMPK activation and disinhibition of mTORC1 signaling are mechanisms accountable for the deficit in protein translation underlying neurodegeneration [52] as well as for the relationship between stimulation of autophagy machinery, MTF treatment and relief from NeP [53]. The sex-dependent assessment of MTF-induced AMPK phosphorylation revealed a lack of AMPK activation in female mice, while at D3, after CCI, we found the highest degree of AMPK activation in male animals, which was still higher at D7. Hence, such potential insensitivity to MTF-induced AMP phosphorylation in female mice may help to understand the absence of long-lasting functional recovery after peripheral nerve injury in female animals. Essentially, while in male mice, MTF treatment induces an early AMPK activation (as also reported with other autophagy inducers), in females showing a hyperactivated AMPK signaling, the MTF seems to induce a paradoxical effect, decreasing both AMPK and SIRT1 phosphorylation [23].

In light of the above results, MTF treatment did not appear to affect the first stage of WD, although we found a strong enhancement of cells in nerve tissue not linked to SC proliferation (cdc2).

Moreover, because of the activation of the immune response, the regenerative capacity of axons after nerve injury and WD is a process largely relying on innate immune cells such as macrophages [54,55]. There is indeed accumulating evidence that macrophages are recruited towards the microenvironment of nerve injury, playing an active and “secretory” role in enabling WD and axon repair by releasing multiple regenerative factors, including cytokines and chemokines [56]. The first striking difference we found in macrophage recruitment was the disparity between non-MTF and MTF-treated mice, irrespective of being male or female animals. MTF administration markedly increased macrophage accumulation at the site of lesion in both sexes, as compared to animals that did not receive MTF treatment. Nevertheless, from a functional point of view, a relevant difference emerged through the comparison between vehicle-treated male and female mice. Indeed, female mice showed a higher level of macrophage accumulation, while macrophages were much lower in male mice. In other terms, although MTF administration increased macrophage accrual in both male and female animals, an impressive amplification of macrophage recruitment after MTF treatment was observed only in male mice. Hence, while macrophage recruitment appears to have a sharp effect on MTF administration in male mice, the macrophage activation in female animals appears to be much less dependent as well as much less sensitive to MTF treatment. On the other hand, this result may be expected in consideration that not only SCs are metabolically sensitive (i.e., provided with the sensor-like AMPK function and mTOR), but macrophages can also change activity upon their exposure to metabolic challenges [57]. Indeed, the activation of the mTOR pathway induces macrophage polarization [58], and MTF is able to switch the macrophage phenotype and regulate macrophage function, including proliferation and differentiation [59,60]. In this view, it is important to consider the role of pro- or anti-inflammatory agents present in the tissue microenvironment.

Among the elevated number of cytokines and agents associated with peripheral nerve damage, pro-inflammatory processes, necrosis and development of NeP, the expression of TNF-α is recognized to play a pivotal role in pain sensitization and a valid potential target in drug discovery aimed at clinical pain management [61,62]. Since its first characterization [63], the production of TNF-α has been viewed as tightly intertwined with the state of activation of immune cells such as macrophages. Accordingly, TNF-α is found upregulated at the site of nerve injury and highly expressed in macrophages [64,65]. In line with these reports, we detected a substantial increase of TNF-α expression in male and female animals that was particularly upregulated at D7 from CCI in comparison to vehicle-treated mice. However, as for the expression of macrophages in vehicle-treated female mice, also TNF-α levels were higher (at D7) in female mice than in male animals. In other terms, considering the lower levels of macrophages and TNF-α expression in non-MTF-treated male mice, the MTF treatment produced a much higher increase of macrophages and TNF-α expression in male than in female mice.

However, as a key mediator of pain processing and pain perception, TNF-α signaling may have a dual role depending on TNF receptors (TNFRs), which include the constitutively expressed TNFR1 and the inducible TNFR2 [66]. Such a dual role has been, for instance, shown by the lack of mechanical allodynia and hyperalgesia in TNFR1 KO mice [67,68], thus supporting the idea that TNFR1 is involved in the development and persistence of NeP. Although still controversial, the prevalent view is that TNFR1 signaling mediates inflammation and apoptosis, while TNFR2 signaling may provide immune regulation, cytoprotection and facilitation of pro-survival pathways [66]. The study of the different temporal profiles of TNFR1 vs. TNFR2 activation during WD disclosed the long-lasting upregulation of TNFR2 up to 28 days after peripheral nerve injury when nerve regeneration occurred, in contrast with the highest activation of TNFR1 during the first days after CCI when maximal is the hyperalgesic response [69]. Emblematically, drug discovery focused on the role of TNF-α in NeP has been focalized either on the blockade of TNFR1 signaling (e.g., in rheumatoid arthritis) [70] or on the design of effective agonists at the TNFR2, which have been used for multiple clinical conditions such as spinal cord injury-induced locomotor deficits [71]. Interestingly, the dual role of TNFRs in inflammation, immune modulation, tissue repairing and pain processing is also mirrored by a different impact on pain sensitivity and recovery among male and female subjects. Indeed, the inhibition of TNFR1 signaling accelerates recovery from NeP in male animals only, an inhibition of therapeutic response that in females was shown to be dependent on estrogen secretion [72]. The same study confirmed that NeP development appears less dependent on TNFR1 signaling in females than in male animals [72]. Our data showed a marked increase of TNFR1 expression at D7 after CCI in male mice, which was not changed by MTF administration. Remarkably, we found a major decrease in TNFR2 expression in female mice that underwent peripheral nerve injury, which seems to support the key importance of TNFR2 signaling in pain modulation and recovery from chronic pain. At difference with MTF-treated male mice, MTF-treated female mice exhibited an initial functional recovery of the allodynic response (i.e., increase of pain threshold) that was later replaced by the loss of initial recovery and aggravation of NeP over a long-time period. Thus, despite the MTF treatment, chronic NeP was not resolved in female animals. Interestingly, a recent study [60] has demonstrated the failure of chronic pain resolution in TNFR2 KO mice. Although the suppression of pain recovery in TNFR2 KO mice was sex-independent [73], this study corroborates the idea that TNFR2 is critical for pain recovery. In our study, TNFR2 expression was drastically reduced only in female mice, and only female mice showed a lack of long-term chronic pain resolution.

Moreover, the regulation of the inflammatory cascade may also contribute to accounting for sex differences in the recovery from CCI-induced allodynia and NeP.

Interestingly, in CCI MTF-treated male mice, we observed an overall decrease in pro-inflammatory interleukins such as IL-1b, IL-2, IL-3, and IL-6, while emerged the selective upregulation of several chemoattractant cytokines such as LIX (CXCL5), Lymphotactin, MCP-1 (CCL2), MIG (CXCL9) and TECK, a framework that can be associated to the regulation of neutrophil and monocyte/macrophages trafficking (e.g., migration) in accordance with previous evidence [59,74]. On the contrary, in female CCI MTF-treated mice, an overall cytokines and chemokines increase was revealed. Higher macrophage expression in female mice that was not reduced but boosted by MTF treatment. This paradoxical increase of pro-inflammatory factors under MTF treatment could explain the different macrophage polarization, which can differently affect pain response and nerve recovery. Given the secretory activity of macrophages, after peripheral nerve damage and activation of tissue-resident macrophages, several soluble pro-inflammatory cytokines and chemokines are generally produced [75]. In other words, the excessive immune system “pre-activation” in female mice that underwent traumatic nerve injury is further enhanced (e.g., exacerbated) in these animals by MTF treatment, thus producing an excessive macrophage accumulation in injured nerves and a disproportionate release of inflammatory factors that disrupted the process of full recovery of hyperalgesia and allodynic response.

Another distinguishing difference between male and female mice is the upregulation of leptin expression, which was observed only in the latter group. Leptin involvement in NeP development is a relatively young yet well-established issue, whose mechanisms are, for instance, linked to the enhancement of NMDA receptor-mediated excitability and/or to the upregulation of the NMDA receptors in the peripheral nervous system (i.e., spinal cord) [76,77]. The investigation of additional mechanisms involved in the leptin-induced development of NeP has identified the important responsibility of this adipocytokine in both microglia activation and proliferation at the spinal cord level and in the brainstem [78]. Sex differences in the impact produced by leptin on NeP are barely studied, although there is evidence that macrophage stimulation facilitates leptin-associated NeP development [79]. Our study disclosed both higher macrophage expression and leptin upregulation in female mice that underwent CCI, also highlighting the fact that MTF treatment did not offset the alteration of either leptin expression or macrophage activation. Accordingly, the cytokine IL-12/p70 and macrophage colony-stimulating factor (M-CSF) were both upregulated in CCI female mice regardless of MTF treatment. Moreover, leptin is known to be involved in the regulation of both innate and adaptive immunity and, therefore, in the modulation of the inflammatory response [80]. Indeed, by activating monocyte proliferation, leptin can enhance macrophage-dependent phagocytic activity as well as the production of pro-inflammatory cytokines such as IL-12 [80].

Finally, our results demonstrate that MTF can have different analgesic potentials in male and female neuropathic mice and that its anti-inflammatory/analgesic action is mediated differently from macrophages rather than exerting a pro-autophagic action on SCs. Altogether our data highlight the importance of taking into account in preclinical and clinical studies the importance of gender differences, especially in consideration of the rather different and dimorphic pharmacological response that could be generated.

## 4. Materials and Methods

### 4.1. Animals

CD1 male and female mice, about 4 (4M) months old, from Charles River Labs (Como, Italy) or EMMA infrafrontier (Monterotondo, RM, Italy) were used. Animals were housed in standard transparent plastic cages, in groups of 4, lined with sawdust under a standard 12/12-h light/dark cycle (07:00 AM/07:00 PM), with food and water available ad libitum. Testing was performed blindly for the treatment group to which each subject belonged. After behavioral testing, the estrous cycle was analyzed in females by means of vaginal smears. Because we did not observe any difference in the behavioral responses, we included all females in the same experimental group independently from the estrous cycle. All procedures were in strict accordance with the European and Italian National law (DLGs n. 26 of 4 March 2014, application of the European Communities Council Directive 2010/63/UE) on the use of animals for research and with the guidelines of the Committee for Research and Ethical Issues of IASP [81]. The number of experimental protocol Ministry of Health aut. n° 32/2014-PR.

### 4.2. Surgery

Following the procedure originally proposed by Bennett and Xie adapted to the mouse, the Chronic Constriction Injury (CCI) model was used as a model of NeP [82]. CCI of the sciatic nerve was performed under anesthesia with a mixture 1:1 of Rompun (xylazine, 20 mg/mL, 0.5 mL/kg, Bayer) and Zoletil (tiletamine and zolazepam, 100 mg/ mL, 0.5 mL/kg); the middle third of the right sciatic nerve was exposed through a 1.5 cm longitudinal skin incision. Three ligatures (7–0 chromic gut, Ethicon, Rome, Italy) were tied loosely around the sciatic nerve. The wound was then closed with a 4–0 silk suture. In the following, the injured right hind paw will be named the ipsilateral paw, and the uninjured left hind paw will be named the contralateral paw.

### 4.3. Drugs

Animals were treated from the day of surgery for 7 days (D7) with an intraperitoneal injection (one a day) of vehicle (saline) or metformin (MTF; metformin chlorhydrate TEVA, Italy) 200 mg/kg dissolved in 0.9% saline.

The experimental timeline is reported in Figure 5.

### 4.4. Allodynia Assessment

Allodynia (a painful response to a non-painful stimulus) was assessed by the measurement of the mechanical nociceptive threshold. Before allodynia assessment, mice were left for 30′ in the experimental room for environmental habituation and in the experimental apparatus for 5′ before testing. The onset of neuropathy was evaluated by measuring the sensitivity of both ipsilateral and contralateral hind paws to normally non-noxious punctuate mechanical stimuli at different time intervals from postoperative day 3 (D3) up to day 60 (D60). The nerve injury-induced mechanical allodynia was tested by using a Dynamic Plantar Aesthesiometer (Ugo Basile, Model 37400, Gemonio VA, Italy), an apparatus that generates a mechanical force linearly increasing with time. The force is applied to the plantar surface of the mouse’s hind paw, and the nociceptive threshold is defined as the force, in grams, at which the mouse withdraws its paw. On each day of testing, the mechanical withdrawal response of ipsilateral and contralateral hind paws was recorded for 3 consecutive trials with at least 10 s between each trial. The withdrawal threshold was taken to be the mean of the 3 trials. The behavioral test was performed in male and female mice (n = 9/10 per group).

### 4.5. Body Temperature

Body temperature (BT) was determined rectally by means of a digital thermometer with an accuracy of 1/10 of a centigrade (°C). The measurement was performed at BL condition and at D7 post-CCI n all experimental groups (n = 9/10 per group).

### 4.6. Glycemia and Triglycerides Measurement

Blood glucose and triglycerides were measured using a Multicare Test Strips apparatus (Biochemical Systems International) by tail clipping in naïve animals (BL) and at D7 after CCI in male and female mice (n = 9/10 per group)

### 4.7. Enzyme-Linked Immunosorbent Assay (ELISA) for TNFα, VIT B12 and SIRT1

Male and female mice concentrations of TNFα and VIT B12 were measured in the serum using an ELISA kit (Single-Analyte ELISArray Kit, Qiagen and Mouse Vitamin B12 ELISA Kit, MyBioSource), while SIRT1 ELISA analysis was performed on male and female mice sciatic nerves homogenates (Mouse SimpleStep SIRT1 ELISA Kit, Abcam) according to the manufacturer’s recommendations. Serum samples and sciatic nerves were harvested at 6 h, 24 h, 3 (D3) and 7 (D7) days after CCI after saline and metformin treatment (at least n = 3 per group/time point). Samples, including standards of known mouse TNFα, VIT B12 and SIRT1, were added to the wells. Following incubations and washes were performed according to the manufacturer’s instructions. The intensity of the colored product was directly proportional to the concentration of TNFα, VIT B12 and SIRT1 present in the blood specimen and homogenates and was read at 450 nm (with a wavelength correction at 570 nm only for TNFα and SIRT1).

### 4.8. Inflammatory Antibody Array

The expression levels of various cytokines/chemokines in the sciatic nerves tissue lysates of male and female mice were analyzed using a mouse antibody array glass chip (RayBio^®^ Mouse Cytokine Antibody Array G series; RayBiotech Inc., Norcross, GA, USA). Lysis buffer (Raybiotech, Inc) containing proteinase inhibitor (Sigma Aldrich) was added to homogenate the sciatic nerves, protein concentration was determined, and 30 µg of each sample was added to the array (at least n = 3 per group). Incubation and washes were performed according to the manufacturer’s instructions. Briefly, chip arrays were blocked at room temperature for 30 min before being incubated with 100 µL of each sample overnight at 4 °C. Glass chips were then washed and incubated with biotin-conjugated primary antibody and fluorescent dye-conjugated streptavidin according to the manufacturer’s instructions. Fluorescence detection was performed using an Agilent G2564B microarray scanner (Agilent Technologies Italy), and data extraction was performed using the array testing services from RayBiotech Norcross, GA, USA.

### 4.9. Immunohistochemical Analysis

The sciatic nerve of mice belonging to each experimental group (at least n = 3/group) was harvested for IF analysis. Animals were sacrificed with a sub-lethal dose of a Rompun (Bayer SpA, Italy) and Zoletil (Virbac Srl, Italy) mixture and perfused with saline followed by 4% paraformaldehyde in phosphate buffer saline (PBS, pH 7.4). The sciatic nerve was removed and kept in immersion for 48 h in 4% paraformaldehyde in phosphate buffer saline (PBS, pH 7.4) after cryoprotection with a solution of 30% (*w*/*v*) sucrose in PBS and maintained at −80 °C. Cryostat microtome sections (20 microns) were taken and mounted directly on glass slides. IF analysis was made before (naïve nerve) and seven days after CCI (D7) in both male and female adult mice. For double IF staining, sections were incubated overnight with the following: (i) anti-S100beta (Schwann cell marker) antibody (mouse monoclonal, 1:100, Sigma-Aldrich: S2532); (ii) anti-CD11b (complement receptor 3/cluster of differentiation 11b, macrophages marker) antibody (rat anti-mouse, 1:100, Millipore: MCA711); (iii) anti-myelin protein zero (P0 or MPZ, myelin marker) antibody (chicken polyclonal, 1:200, AB9352; Millipore); (iv) anti-peripheral myelin protein 22 (PMP22) antibody (rabbit polyclonal 1:200 Sigma-Aldrich, sab4502217); neurofilament 200 (NF200, myelinated axons markers) antibody (rabbit polyclonal 1:100, Sigma-Aldrich N4142.); anti-cyclin-dependent kinase (Cdc2, proliferation marker) antibody (rabbit polyclonal 1:100, Calbiochem, PC25). All antibodies were diluted in Triton 0.3% (Sigma-Aldrich St. Louis, MO, USA).

After three washes in PBS, sciatic nerve sections were incubated for 2 h at room temperature with fluorescein-conjugated donkey anti-mouse (ALEXA Fluor 488, 1:100, Jackson ImmunoResearch), fluorescein-conjugated rat anti-mouse (FITC, 1:100, Jackson ImmunoResearch) or rhodamine-conjugated donkey anti-chicken DyLight 549 (DYL, 1:100, Jackson Immuno Research), goat anti-rabbit IgG-FITC (1:100 Santa Cruz, sc-2012), or rhodamine (TRICT) goat anti-rabbit (1:100, Jackson ImmunoResearch) secondary antibodies in 0.3% Triton. After 2 washings in PBS, sections were incubated for 10 min with bisBenzimide, DNA-fluorochrome (Hoechst, 1:1000, Sigma-Aldrich) in PBS.

To exclude nonspecific signals of secondary antibodies and to warrant optimal results, both control and treated sections have also been stained with secondary antibodies alone (negative control).

### 4.10. Confocal Images and Analysis

Images of the immunostained sections were obtained by laser scanning confocal microscopy using a TCS SP5 microscope (Leica Microsystems). All analyses were performed in sequential scanning mode to rule out cross-bleeding between channels. High magnification (63×) images of sciatic nerve sections were operated by I.A.S. software (Leica Microsystems Srl, Milan, Italy). Quantification was performed by using the ImageJ software (version 1.41, National Institutes of Health, Bethesda, MD, USA). The fluorescence of proteins observed was quantified (at least 2 slices × n = 3 each group) by converting pixels in brightness values using the RGB (red, green and blue) method that is largely applied to detect, digitalize and analyze microscopic images from biological samples by a confocal microscope [37] and by subtracting for each image the background portion of the analysis. Macrophages and nuclei quantification was performed by using the ImageJ software (version 1.41; National Institutes of Health, Bethesda, MD, USA), automatically counting the number of IF-positive cells (CD11b or Hoechst) by means of the mark and count tool, and then the mean for each group of mice was calculated.

### 4.11. Western Blot Analysis for pAMPK

For pAMPKα analysis, a total of 50 µg of sciatic nerves (pool of 3 nerves from 3 different animals for each single experiment, and 50 ug of total protein lysate for each gel was loaded), from 3 experimental groups (naïve, saline and metformin), at four time points post CCI (6 h-24 h-3 days-7 days) were used (n = 3 mice/group/time point). Membranes were incubated overnight at 4 °C with the following primary antibody: pAMPKα (Phospho-AMPKα-T172 polyclonal antibody; Elabscience) and β-actin (AM1829b, beta Actin, monoclonal antibody; Abgent). Antibody binding was revealed by using enhanced chemiluminescence (ECL) (Euroclone, Pero, Mi, Italy). Luminescent bands were imaged with autoradiography (X-ray) films (UltraCruz; Santa Cruz Biotechnology, Dallas, TX, USA) and then scanned into a digital format. The β-actin band intensities were used as a control for equal protein loading and measured for densitometric analysis (AMPK/Actin) using ImageJ 1.49r software (National Institutes of Health, USA).

### 4.12. Statistics

All values are expressed as mean ± SEM. The homoscedasticity assumption was verified by the Levene test. The sample size, relative to the in vivo experiments, was previously calculated by implementing a Power analysis (GPower 3.1). For the ex-vivo experiments, the sample size is based on our previous experience and in agreement with animal ethics and the 3 “R” principles. Depending on the data, statistical analysis was performed either by unpaired t-test, 1-way analysis of variance (ANOVA) or 2-way ANOVA for repeated measures, while for small samples (n < 5 animals) and groups > 3, non-parametric analysis was performed by Kruskal–Wallis. Tukey–Kramer test has been used for post hoc analysis in multiple comparisons or t-tests for single comparisons. Differences were considered significant at *p* < 0.05. For statistics, Statview 5.0 and Rstudio were utilized.

## 5. Conclusions

Altogether, these data outline a different “sex-dependent” response following the use of MTF as an analgesic drug for the treatment of NeP. From our study, it emerges that MTF repurposing for its antinociceptive potential should be considered cautiously, and in particular that greater attention should be paid to the caveat of the lack of durable antinociceptive effects and loss of analgesic effects in female individuals.

## Figures and Tables

**Figure 1 ijms-23-14503-f001:**
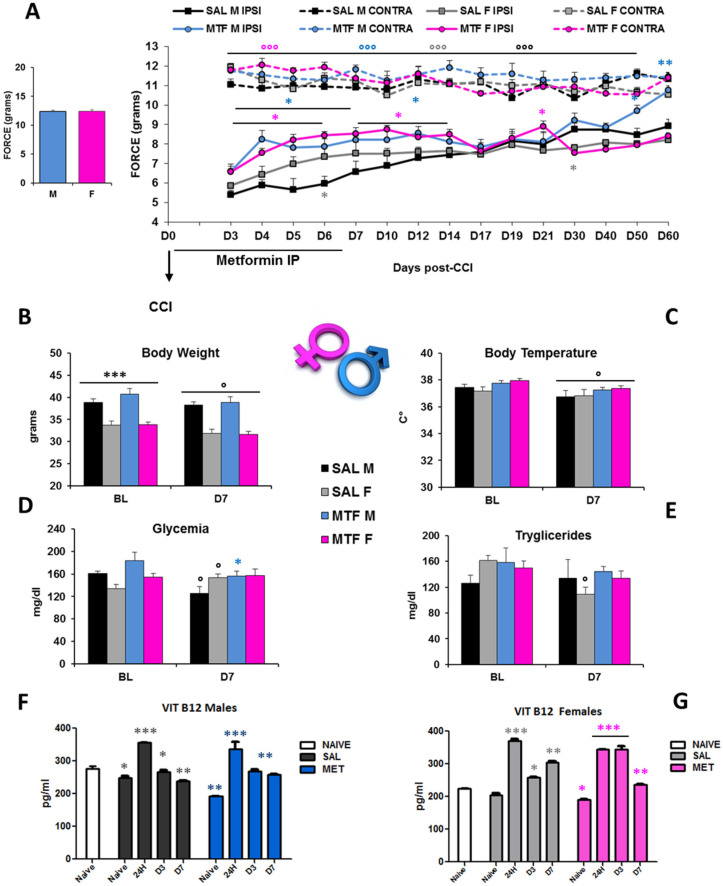
Sex-dependent anti-allodynic effects of MTF administration and metabolic changes. (**A**) left graph: threshold of response (force in g) to non-noxious stimuli (dynamic plantar aesthesiometer test) in no-lesioned mice (baseline-BL); right graph: Allodynic response measured from day 3 to day 60 in subchronic (from D1 to D7) vehicle (saline-SAL) or metformin (MTF) treated male (M) and female (F) CD1 sciatic nerve-injured (CCI) mice. Dotted lines refer to uninjured contralateral (CONTRA) hind paw; solid lines refer to injured ipsilateral (IPSI) hind paw (n = 9/10 each group; ANOVA repeated measures: °°° IPSI vs. CONTRA F_7,70_ 127,619 *p* < 0.0001; Time F_70,14_ 7008 *p* < 0.0001; treatment × time F_98,980_ 4.444 *p* < 0.0001; t-student SAL M vs. SAL F * grey *p* < 0.05; MTF M vs. SAL M * blue *p* < 0.05; ** *p* < 0.001; MTF F vs. SAL F * pink *p* < 0.05). Metabolic changes produced by neuropathy and MTF treatment on body weight (**B**) *** *p* < 0.0001 males vs. females; ° *p* <0.05 D7 vs. BL; (**C**) body temperature ° *p* < 0.05 D7 vs. BL; (**D**) glycemia ° *p* < 0.05 D7 vs. BL, * *p* < 0.05 MTF vs. SAL; (**E**) triglycerides ° *p* < 0.05 D7 vs. BL-in vivo analysis (n = 9/10 group); and (**F**) vitamin B12 levels in males and (**G**) females; Unpaired T-test for comparison * *p* < 0.05; ** *p* < 0.001; *** *p* < 0.0001 vs. naïve (ex-vivo analysis blood levels-ELISA; n = 3 each group; the Kruskal–Wallis test H_8,9_ 23.2 *p*: 0.0031 (males); H_8,9_ 25.38 *p*: 0.0013 (females)).

**Figure 2 ijms-23-14503-f002:**
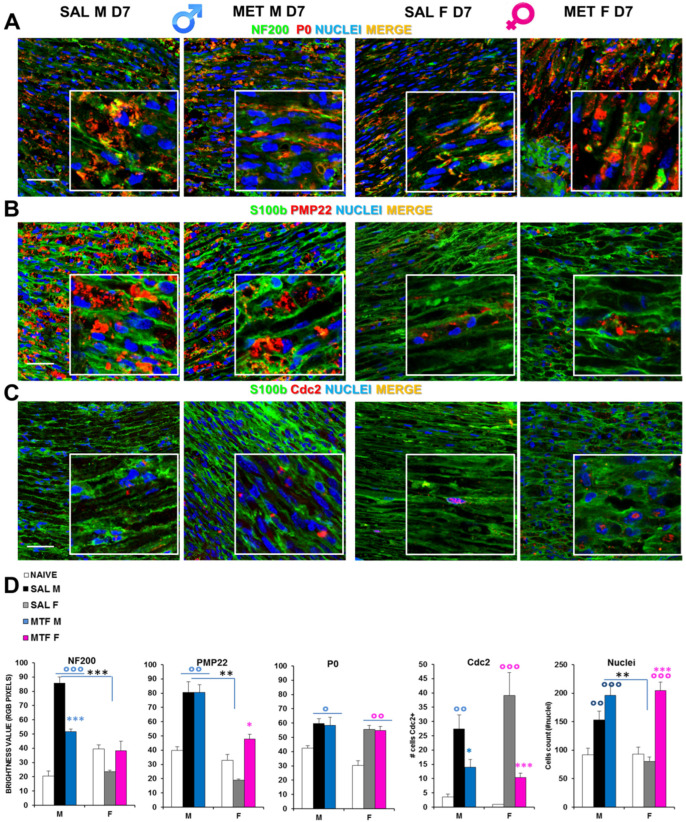
Effects of MTF administration on CCI-induced peripheral (sciatic nerve) neurodegeneration in male and female mice. (**A**) Representative merged confocal images (high magnification 63× and zoom 2×; scale bar 50 u) of the lesioned area of the sciatic nerve stained for neurofilament 200 (NF200, green), myelin protein zero (P0, red) and nuclei (cyan); or for (**B**) Schwann cell marker, S100beta (green), peripheral myelin protein 22 (PMP22, red); or for (**C**) S100b (green) and Cell Division Cycle 2 (Cdc2, red); colocalizing proteins are yellow. (**D**) Graphs show RGB (red, green, blue) analysis transforming pixels in brightness values for NF200, P0 and PMP22 markers; the number of cells positive for Cdc2 and the total number of nuclei were counted. (n = 3/5 animals/group, 2/3 slices/animal) °, °°, °°° (blue or pink): *p* < 0.05, 0.005, 0.0001 respectively vs. CTRL males or females; *, **, *** (blue or pink): *p* < 0.05, 0.005, 0.0001 respectively vs. saline (SAL) males or females; **, *** (black): *p* < 0.005 and 0.0001 SAL males vs. SAL females. ANOVA for NF200:F_5,40_: 25.46 *p* < 0.0001; P0: F_5,45_: 5104 *p* < 0.0001; PMP22 F_5,40_ 18,196 *p* 0.0010; Cdc2 F_5,30_ 9919 *p* < 0.0001; Nuclei F_5,104_ 13,827 *p* < 0.0001.

**Figure 3 ijms-23-14503-f003:**
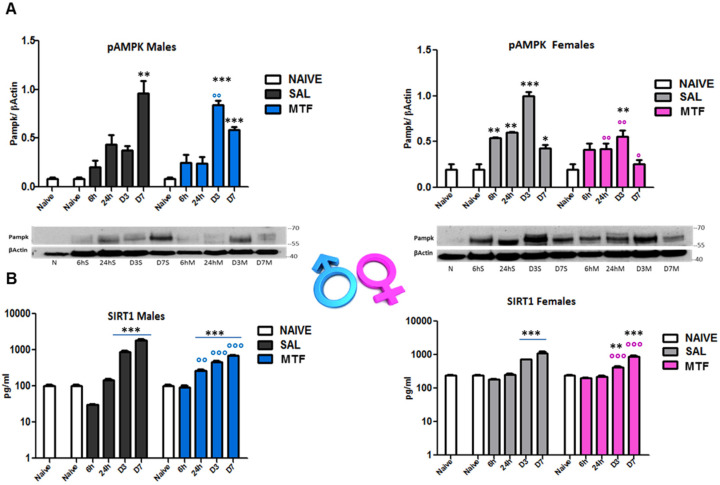
Effects of MTF administration on pAMPK and SIRT1 activation in the injured sciatic nerve. (**A**) pAMPK western blot analysis of male and female sciatic nerve lysates of naive sciatic nerves and nerves evaluated at different time points from the lesion (6 h–6 h, 24 h, day 3–D3, D7). (**B**) SIRT1 ELISA analysis of sciatic nerve lysates of naive sciatic nerves and nerves evaluated at different time points from the lesion (6 h–6 h, 24 h, day 3–D3, D7). T-student comparison: *, **, *** *p* < 0.05, 0.005, 0.0001 respectively vs. naive; °, °°, °°° (blue or pink) *p* < 0.05, 0.005, 0.0001 respectively vs. saline (SAL) males or females; n =3/4 animals/group. Kruskal–Wallis: pAMPK (males) H_8,9_: 21,73 *p*: 0.0054; pAMPK (females) H_8,9_: 22,529 *p* = 0.004; SIRT1 (males) H_9,10_: 25.59 *p* = 0.0024; SIRT1 (males); SIRT1 (females) H_12,13_: 31,367 *p* = 0.0017.

**Figure 4 ijms-23-14503-f004:**
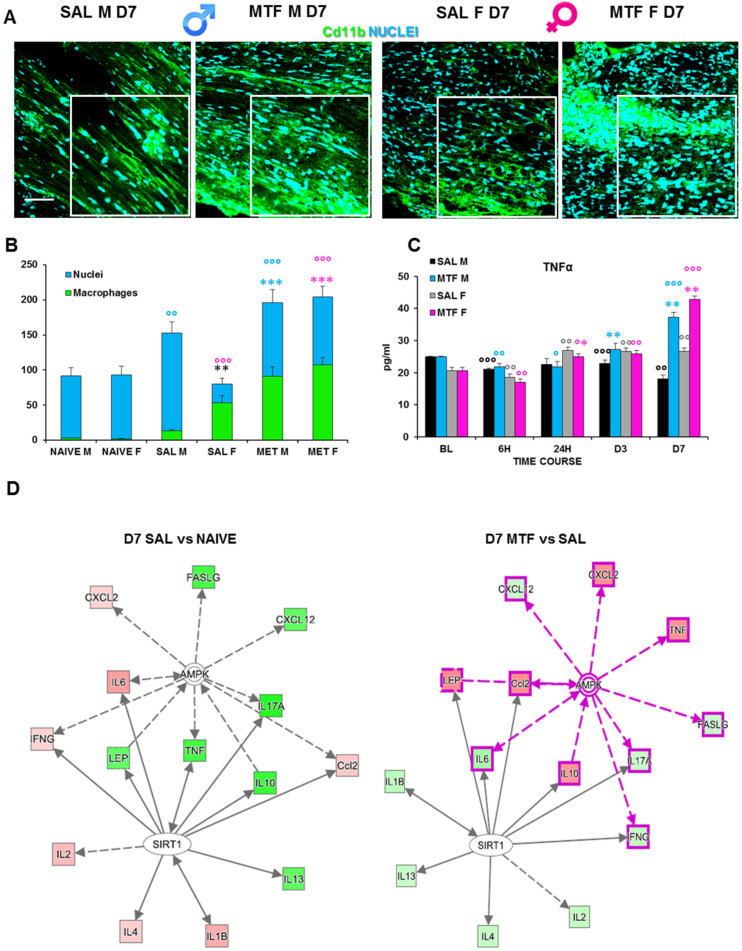
Effects of MTF administration on macrophages at D7 after peripheral nerve injury in male and female mice. (**A**) Sample of confocal images (high magnification 63×, in the selected square zoom 1×, scale bar 50 u) of CD11b (macrophages cell marker, green) positive cells (cyan nuclei) in lesioned sciatic nerves (7 days after CCI) of male and female mice. (**B**) Cd11b cells count (n = 3 animals/group 2/3 slices/animals—H_5,6_: 29,080 *p*: < 0.0001); T-test comparison: ** (black) vs. SAL M *p* < 0.005; *** (blue or pink) vs. SAL M or F respectively *p* < 0.0001; °°°, °° (blue or pink) vs. naive M or F respectively *p* < 0.005 and *p* < 0.0001. (**C**) Time-course (BL-baseline, 6 h–6 h, 24 h, day 3–D3 and D7 after CCI) of TNFα levels in sciatic nerve lysates of male and female mice. (n = 3/group/time-point) males: H_8,9_: 21,6 *p* = 0.0057; females: H_8,9_: 26,527 *p* < 0.0009; Unpaired t-test comparison: °, °°, °°° vs. BL *p* < 0.05, 0.005, 0.0001 respectively; *, **, *** vs. SAL *p* < 0.05, 0.005, 0.0001 (**D**) Schematic representation deriving from Ingenuity Pathway Analysis of molecular interactions and networks of proteins found variated (red, upregulated and green, downregulated) in females and males 7 days after nerve ligature (left, gray-gray network: saline vs. naive; right, pink-gray network: metformin vs. saline). A solid line represents a direct interaction between the two gene products, and a dotted line means there is an indirect interaction.

**Figure 5 ijms-23-14503-f005:**
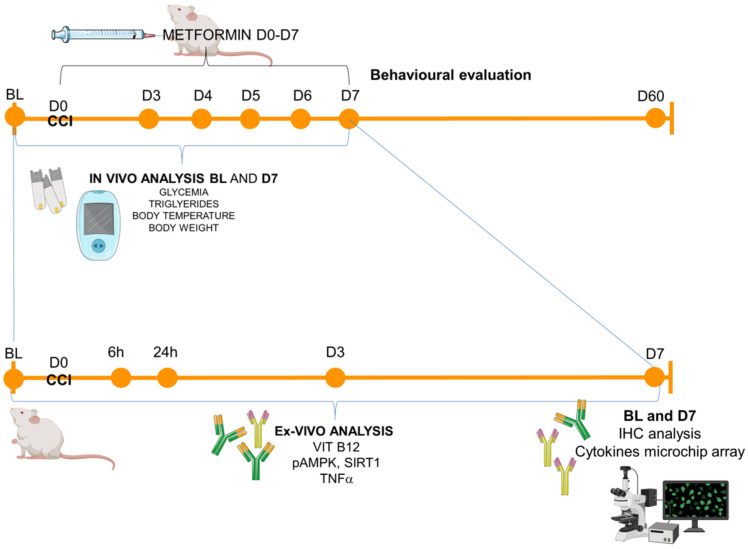
Experimental timeline. Schematic representation of the experimental schedule: animals (male and female CD1 mice) were tested in the baseline condition (BL) before the surgical induction of neuropathy (CCI–day 0, D0) and starting from day 3 (D3) post-ligature up to day 60 (D60). In-vivo metabolic measurements were performed in BL and D7, while for the ex-vivo analysis (from blood or sciatic nerves), the following time points were considered: BL (naïve mice), 6 h, 24 h, D3, D7 for ELISA immunoassay (Vit B12, pAMPK, SIRT1, TNFa) and BL and D7 for immunohistochemistry (IHC) and microchip array inflammatory panel.

**Table 1 ijms-23-14503-t001:** Cytokines Changes at D7 After CCI in Naïve and Injured Sciatic Nerve of Saline or MTF-Treated Male and Female Mice.

		FI VALUES			FI VALUES		
MEDIATOR	NAME and FUNCTION	NAIVE M	SAL CCI D7 M	MTF CCI D7 M	NAIVE F	SAL CCI D7 F	MTF CCI D7 F
**BLC**	B lymphocyte chemoattractant CXCL13	3441.48 ± 4.05	3098.05 ± 55.51	3146.89 ± 92.62	30.11 ± 5.13	21.29 ± 1.52	**18.78 ± * 0.70**
**CD30L**	tumor necrosis factor ligand superfamily member 8	3239.11 ± 145.84	3202.57 ± 178.68	2888.45 ± 99.60	8.16 ± 0.67	**50.85 ± ** 1.16**	**49.84 ± ** 3.70**
**Eotaxin-1**	C-C motif chemokine 11; eosinophil chemotactic protein	44,572.06 ± 558.62	**86,503.33 ± ** 1113.81**	**69,609.32 ± * ° 244.64**	1136.88 ± 91.29	**8543.73 ± ** 203.06**	**7010.47 ± ** 254.25**
**Eotaxin-2**	CC chemokine selective for the chemokine receptor CCR3	65,439.39 ± 707.19	62,086.92 ± 1392.43	**35,915.51 ± ° 1376.24**	368.33 ± 22.90	**920.55 ± ** 6.64**	**1462.87 ± ** ° 33.42**
**FAS ligand**	type-II transmembrane protein tumor necrosis factor (TNF) family; apoptosis	2678.89 ± 419.10	2188.06 ± 19.53	2033.72 ± 78.96	5.58 ± 0.24	**17.40 ± ** 1.02**	**17.86 ± ** 0.11**
**Fractalkine**	chemoattractant activity for T cells and monocytes	2800.60 ± 450.10	2187.31 ± 114.32	2132.81 ± 29.68	28.50 ± 0.58	30.97 ± 4.12	33.73 ± 1.73
**G-CSF**	Granulocyte colony-stimulating factor	4717.01 ± 622.37	**9196.65 ± ** 4132.10**	**2200.45 ± * ° 45.01**	19.01 ± 2.01	**152.38 ± ** 5.10**	**125.27 ± ** 10.62**
**GM-CSF**	Granulocyte-Macrophage Colony-Stimulating Factor	3334.68 ± 9.71	2918.11 ± 6.60	3036.82 ± 81.25	48.21 ± 3.63	62.51 ± 2.81	**106.72 ± ** ° 21.33**
**IFN-gamma**	Interferon gamma (IFN-γ) innate and adaptive immunity, primary activator of macrophages,	3371.72 ± 6.77	3506.79 ± 139.52	2975.81 ± 20.65	79.93 ± 6.10	83.36 ± 2.35	93.88 ± 3.77
**IL1-alpha**	interleukine, inflammtory cytokine	2764.76 ± 29.02	3433.61 ± 397.70	3396.78 ± 114.17	18.42 ± 4.66	23.38 ± 0.36	**32.55 ± * ° 1.61**
**IL1-beta**	interleukine, inflammtory cytokine	2545.66 ± 203.90	**4826.89 ± ** 877.03**	**3726.47 ± * 551.78**	11.98 ± 1.15	19.40 ± 0.77	**25.92 ± * 0.60**
**IL2**	interleukine activateing cytotoxic T cells and NK cells	3680.88 ± 28.07	**5848.85 ± * 758.01**	**5081.43 ± * 349.40**	65.87 ± 6.45	82.10 ± 7.06	**94.64 ± * 8.43**
**IL3**	interleukine, multicolony-stimulating factor	3956.59 ± 12.65	**5625.25 ± * 673.60**	4897.04 ± 501.71	67.37 ± 0.65	**104.35 ± * 4.68**	**107.91 ± * 4.26**
**IL4**	interleukine, prototypic immunoregulatory cytokine.	4695.06 ± 21.11	5408.58 ± 400.68	4864.79 ± 358.95	75.66 ± 4.43	**111.38 ± * 4.51**	**110.39 ± * 2.03**
**IL6**	pro-inflammatory cytokine and an anti-inflammatory myokine	3126.88 ± 57.67	**6946.52 ** ± 302.31**	4110.4 ± 154.86	73.17 ± 6.16	**98.85 ± * 5.46**	**90.50 ± * 0.71**
**IL9**	interleukine, T cell growth factor	4778.08 ± 25.16	4012.32 ± 61.27	4494.81 ± 85.48	88.53 ± 7.48	**124.79 ± * 1.26**	**130.53 ± * 1.27**
**IL10**	interleukine, cytokine synthesis inhibitory factor (CSIF), anti-inflammatory cytokine	3486.74 ± 285.06	**2664.39 ± * 81.40**	2807.9 ± 37.64	47.91 ± 4.95	**74.77 ± * 1.71**	**76.21 ± * 0.51**
**IL12-p40/p70**	produced mainly by macrophages, induction of NK cells, elaboration of IFN-γ,	3365.14 ± 469.47	2461.49 ± 68.61	2718.25 ± 115.26	8.13 ± 0.21	**44.193 ± ** 1.27**	**40.79 ± ** 1.81**
**IL12-p70**	produced mainly by macrophages, induction of NK cells, elaboration of IFN-γ,	5636.57 ± 282.94	4482.43 ± 103.21	4495.88 ± 51.06	116.27 ± 9.97	124.30 ± 6.33	138.67 ± 3.14
**IL13**	immunoregulatory cytokine, regulating function of human B cells and monocytes (but only macrophages in the mouse).	2310.25 ± 35.07	2216.17 ± 19.33	2086.64 ± 46.34	1.52 ± 0.15	**17.17 ± ** 1.45**	**23.66 ± ** 3.52**
**IL17**	interleukin, links T cell activation to neutrophil mobilization and activation	4972.98 ± 588.04	**3400.63 ± * 34.76**	**3315.95 ± * 78.54**	116.98 ± 8.67	106.50 ± 7.33	124.79 ± 4.90
**I-TAC**	CXCL11, interferon-inducible T cell alpha chemoattractant	3499.39 ± 193.76	**2975.77 ± * 122.78**	**2868.62 ± * 151.32**	7.45 ± 2.57	11.06 ± 0.10	5.91 ± 0.88
**KC**	keratinocytes-derived chemokine	2307.15 ± 84.95	2361.44 ± 32.06	2595.65 ± 109.03	10.47 ± 1.43	12.05 ± 0.37	16.91 ± 0.10
**Leptin**	Hormone, increasing the cytotoxicity of natural killer (NK) cells, activation of granulocytes, macrophages	2236.96 ± 76.20	2199.3 ± 25.46	2545.42 ± 141.55	7.39 ± 0.81	**17.55 ± ** 0.86**	**14.77 ± ** 0.21**
**LIX**	Chemokine (C-X-C motif) ligand 5 (CXCL5), induced IL-1beta and TNF-alpha promoter activity	3121.82 ± 80.72	3295.84 ± 102.97	4746.27 ± 696.05	21.53 ± 5.79	**57.20 ± ** 2.28**	**54.71 ± ** 4.50**
**Lymphotactin**	chemokine, recruiting T and NK cells, produced by activated CD8+ T-, NK -cells.	2834.72 ± 104.63	2801.68 ± 162.04	**5827.55 ± ** °° 214.64**	75.90 ± 4.67	113.59 ± 3.54	113.92 ± 2.69
**MCP-1**	chemokine, Monocyte Chemoattractant Protein-1	4260.97 ± 261.81	5354.87 ± 198.88	6100.70 ± * 1179.72	117.17 ± 6.16	102.50 ± 3.05	98.42 ± 2.06
**M-CSF**	Macrophage colony-stimulating factor, regulating monocytes proliferation, differentiation, activation	3853.74 ± 51.24	3836.77 ± 61.51	4299.38 ± 81.61	5.60 ± 0.35	**24.59 ± ** 1.03**	**30.92 ± ** 0.67**
**MIG**	Chemokine (C-X-C motif) ligand 9 (CXCL9) recruitment of activated T-cells to sites of infection	2170.33 ± 74.10	2392.14 ± 40.75	3180.47 ± 161.89	14.21 ± 1.03	**32.54 ± * 2.93**	**31.94 ± * 1.00**
**MIP-1-alpha**	Macrophage inflammatory protein-1 alpha	2182.03 ± 109.71	2135.78 ± 96.75	2197.23 ± 62.74	7.52 ± 3.45	10.28 ± 0.55	8.90 ± 2.18
**MIP-1-gamma**	Macrophage inflammatory protein-1 gamma	325,343.87 ± 1176.85	479,722.61 ± 29450.37	473,402.46 ± 9598.91	3762.50 ± 91.22	**12,080.64 ** 679.14**	**10,802.93 ± ** ° 43.48**
**RANTES**	Regulated upon Activation, Normal T Cell Expressed and Presumably Secreted (CCL5)	3256.60 ± 230.71	2755.26 ± 51.61	2867.5 ± 50.26	4.36 ± 1.36	**13.62 ± ** 0.81**	**17.44 ± ** 1.48**
**SDF-1**	stromal cell-derived factor 1 (SDF-1), C-X-C motif chemokine 12 (CXCL12),	2400.66 ± 21.12	2277.58 ± 15.98	2228.74 ± 57.48	21.53 ± 3.12	30.40 ± 0.49	31.21 ± 0.57
**TCA-3**	Activated T lymphocytes (CCL1) orchestrating cellular infiltration during a cell-mediated immune reaction	3364.82 ± 6.25	4225.5 ± 473.93	3496.03 ± 184.16	86.04 ± 5.75	108.40 ± 4.54	112.25 ± 3.10
**TECK**	Thymus-Expressed Chemokine (CCL25)	4015.28 ± 156.62	3477.89 ± 17.08	**5766.24 ± * °° 224.75**	79.31 ± 1.26	**174.88 ± * 7.61**	**249.76 ± ** 16.96**
**TIMP-1**	Tissue inhibitors of metalloproteinases	3027.73 ± 59.31	**14,256.16 ± ** 41.94**	**11,964.84 ± **** 714.31**	7.38 ± 0.22	**478.07 ± ** 24.07**	**460.23 ±** 11.25**
**TIMP-2**	Tissue inhibitors of metalloproteinases	2788.91 ± 114.53	2700.57 ± 39.73	3188.69 ± 263.63	33.98 ± 5.18	**73.65 ± * 5.42**	44.98 ± 0.33
**TNF-alpha**	Tumour Necrosis Factor alpha (TNF alpha), inflammatory cytokine produced by macrophages/monocytes during acute inflammation	3480.15 ± 110.36	2820.75 ± 34.92	3132.01 ± 110.97	114.98 ± 18.18	82.34 ± 2.66	84.18 ± 3.24
**sTNF RI**	Soluble Tumor Necrosis Factor Receptor I	40,114.16 ± 729.10	**104,619.27 ± * 4737.74**	**125,010.95 ± ** 489.83**	618.50 ± 16.74	**1462.48 ± ** 90.02**	**1233.84 ± * ° 29.50**
**sTNF RII**	Soluble Tumor Necrosis Factor Receptor II	8069.29 ± 156.29	**11,281.71 ± * 1115.63**	**5956.14 ± * ° 275.76**	139.88 ± 0.22	165.51 ± 13.87	**120.15 ± ° 9.24**

The table shows cytokines changes analyzed by chip-array in nerve tissue lysates samples. Mean values (three different samples for the group) and standard deviation (SD) of intensity fluorescence (FI), calculated after the chip scan, are reported. For males and females, two different chips were used. In the first, tissue lysates of all the experimental groups from male mice were loaded, and in the second one from all female mice. Significant data are marked with *, ** vs. naïve or °, °° vs. SAL CCI D7 (*p* < 0.05, 0.005, 0.0001 respectively) bolded numbers for upregulated or downregulated mediators. Any ≥ 1.5-fold increase or ≤ 0.65-fold decrease in signal intensity for a single analyte between samples may be considered a measurable and significant difference in expression (as reported by the datasheet).

## Data Availability

The data presented in this study are available on request from the corresponding author.

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
