# Peer review of "Sex Differences in Neuropathy: The Paradigmatic Case of MetFormin"

_ijms, 2022, doi:10.3390/ijms232314503_

Round 1
Reviewer 1 Report (Previous Reviewer 1)
All concerns addressed.
Accept in current form
Author Response
We thank the referee.
Reviewer 2 Report (New Reviewer)
The experimental article is devoted to the hot topic of determining the sex-dependent difference in pain.
Some comments to the authors:
1) please add the timeline of the experiment when you conducted all manipulations day by day (when the tissues were collected for analysis, when metformin was administered etc).
2) long-term use of metformin is generally associated with an increase in neuropathic pain, why do you use metformin in your research to relieve neuropathic pain?
3) why the tissues were taken only on the 7th day and did not look at the more delayed outcomes?
Author Response
We thank the reviewer for the positive comments and productive enquiries addressed to our paper. As suggested, we revised the introduction and we had language checked by an expert mother tongue.
R: 1) please add the timeline of the experiment when you conducted all manipulations day by day (when the tissues were collected for analysis, when metformin was administered etc).
AU: We have now added the experiments timeline in method section, as suggested.
R: 2) long-term use of metformin is generally associated with an increase in neuropathic pain, why do you use metformin in your research to relieve neuropathic pain?
AU: the prolonged use of MTF can induce neuropathic pain (mainly due to alteration in VIT B12 levels that we keep under control during experiments). Nevertheless, although an increasing number of papers suggest MTF as suitable drug for pain treatment, our purpose was to assess the ability of MTF to induce autophagy via AMPK activation and to test the hypothesis that MTF could modulate the first phase of Wallerian degeneration facilitating nerve recovery processes. To better clarify this point, we have now added further information on this topic in the introduction and in discussion.
R: 3) why the tissues were taken only on the 7th day and did not look at the more delayed outcomes?
AU: this is an important point focused on the long-term changes of neuropathy-associated pain induced by an early MTF treatment, and above all in the view of dimorphic response to different drugs as well as to the different metabolic pathways and neuro-immune response differently activated in males and females following nerve injury.
In fact, on this point, we have investigated peripheral and central (spinal cord) effects at 60 days from CCI, and we found (Pain. 2014 Feb;155(2):388-402. doi: 10.1016/j.pain.2013.10.027) a lack of recovery from neuropathy in females as well as different microglial pattern of activation. Thus, it could be of interest to understand if modifying the early (acute) response to nerve injury can interfere with NeP chronicization and nerve restore above all in female mice.
In the present paper, instead, we focused on the first 7 days because we were primarily interested in modulating the myelin/axons debris clearance (autophagic response), mediated by Schwann cell and/or macrophages, which reaches the peak during the first 7 days. We have now better highlighted this point in the introduction and discussion.
This manuscript is a resubmission of an earlier submission. The following is a list of the peer review reports and author responses from that submission.
Round 1
Reviewer 1 Report
Well written, in depth study using appropriate methods. Vast amounts of quality data acquired and properly interpreted.
Minor proofread would be good.
Hypothesis The authors presented current literature and their own research as the basis of the hypothesis that metformin may provide a sex-dependent different degree of analgesia and neuroprotection after CCI-induced allodynia and chronic neuropathic pain. They used male and female mice as a model to test the hypothesis by documenting the temporal evolution of allodynia after administering subchronic metformin and the possible sexual dimorphism in signaling pathways involved in both metformin mechanism of. The methods were appropriate, well performed and the data analyzed correctly The data showed a different “sex-dependent” response following the use of metformin as analgesic drug for the treatment of neuropathic pain. The conclusion is that metformin use should be considered cautiously as a nociceptive treatment especially in females. Although not fatal the authors should address the following questions: 1. Would there be any expected differences if the females were in estrous or gravid or menopausal? 2. How relatable is mouse pain to human pain. It is recognized that pain trials in humans is unlikely. 3. Why are all of the immunochip array antigens concentrations 2 to 10-fold greater in males than females regardless of treatment?
Author Response
R: Well written, in depth study using appropriate methods. Vast amounts of quality data acquired and properly interpreted.
Minor proofread would be good.
Hypothesis The authors presented current literature and their own research as the basis of the hypothesis that metformin may provide a sex-dependent different degree of analgesia and neuroprotection after CCI-induced allodynia and chronic neuropathic pain. They used male and female mice as a model to test the hypothesis by documenting the temporal evolution of allodynia after administering subchronic metformin and the possible sexual dimorphism in signaling pathways involved in both metformin mechanism of. The methods were appropriate, well performed and the data analyzed correctly The data showed a different “sex-dependent” response following the use of metformin as analgesic drug for the treatment of neuropathic pain. The conclusion is that metformin use should be considered cautiously as a nociceptive treatment especially in females. Although not fatal the authors should address the following questions:
- Would there be any expected differences if the females were in estrous or gravid or menopausal?
AU: We thank the reviewer for his/her positive and supportive comments to our work.
Concerning the issue of mice estrous cycle or the state of pregnancy, we agree with the reviewer as for the importance of the matter.
Indeed, in our previous works we have studied if oestrus can affect neuropathic pain response and as demonstrated (Vacca, V.; et al. 17Beta-Estradiol Counteracts Neuropathic Pain: A Behavioural, Immunohistochemical, and Proteomic Investigation on Sex-Related Differences in Mice. Sci. Rep. 2016, 6, doi:10.1038/srep18980; Vacca, V, et al Higher Pain Perception and Lack of Recovery from Neuropathic Pain in Females: A Behavioural, Immunohistochemical, and Proteomic Investigation on Sex-Related Differences in Mice. Pain 2014, 155, doi:10.1016/j.pain.2013.10.027.) no differences were observable in allodynia and hyperalgesia between different phases of the cycle in the same animals or in comparison with other females in the same or in different cycle phase.
Concerning menopausal, our recent data on 12-18 months old mice, not published yet, we observed a pain response to neuropathy that is very similar between males and females, thus confirming the interfering action of sex steroids with neuropathy, pain onset, and maintenance.
R: 2. How relatable is mouse pain to human pain. It is recognized that pain trials in humans is unlikely.
AU: Although mouse and human could be very “far”, pain (also as function of the type of pain: i.e inflammatory, neuropathic, visceral…) per se induces very similar and standardizable/classifiable behavioral response in both species, which is mirrored by the severity and intensity of pain. The decrease of this response is a reliable measure of the efficacy of analgesic drugs. Undoubtedly in humans, thanks to the detailed description of the patients, we have the possibility to explore thoroughly several pain aspects possibly undetected in mice.
With regard to the failure of the most part of drugs present in clinical trials, there are several possible explanations for that. We could even prepare a manuscript on this…
For the sake of brevity, we can detect a general cause underlying the reproducibility of the results (i.e. methods, although these can be precisely described, each lab might obtain different data) hindering the translational research; a second difficulty is related to the intrinsic differences between the species in particular for metabolism and immune system that affect pharmacokinetic and pharmacodynamic as well as toxicity and safety.
This theme is particularly interesting, but statistically only 5% of the compounds is licensed to the clinical phases and in lack of the preclinical research, this percentage would decrease further, thus increasing the risks (above all for patients enrolled in clinical trials).
R: 3. Why are all of the immunochip array antigens concentrations 2 to 10-fold greater in males than females regardless of treatment?
AU: Numbers in the table are referring to the intensity of fluorescence values, calculated after chip scan. For males and females, we have utilized two different chips. One, in which we loaded tissue lysates of all the experimental groups from male mice, and in the second one from all female mice. In fact, we didn’t compare Males vs Females but we discuss only changes in cytokines in comparison to naïve or treatment inside the same sex-group. The difference can be attributable to software detection and technical evaluation (semi-quantitative) on separate experiments. The aspect relevant, in this kind of analysis, is the direction of the variation (up- or down-regulation, one/two…fold).
Reviewer 2 Report
The study by de Angelis et al. aims at examining the role of metformin as a treatment for neuropathic pain induced by CCI in male and female mice. The authors propose that the glucose-lowering medicine may have different efficacies that can be explained by sex-dependent changes in APMK-driven pathways and autophagy. They also examined levels of various phenotypic markers often associated with peripherally driven repair after nerve injury and also levels of chemokines and cytokines that may be involved in this process.
Albeit the topic of neuropathic pain and the so-called gender gap is timely and relevant, this study has many important shortcomings that dampen my enthusiasm for this work.
My main concerns follow, not necessarily in order of importance:
1. Animals: exact number of animals used per group and experiment must be included and justified. Method of blinding (single or double?) must be detailed.
2. Surgery: indicate the drugs rather than the commercial names of the anaesthetic agents used. Include relevant citations for the case of the combination of Tiletamine and Zolazepam having moderate to strong analgesia that lasts far longer than the anaesthesia they induce. This is a potentially contaminating factor when conducting behavioural assessments that are reflex based. Why have the authors omitted a sham operated group as a control?
3. Drugs: i.p. injection of metformin is a highly unusual method of delivery, as this drug is essentially administered orally to almost all patients. Also, i.p. injection is painful and often less well tolerated than the oral administration. The latter has the advantage of replicating far more closely the pharmacodynamics and pharmacokinetics of the drug as it is used in the clinical context. Authors show demonstrate that this form of administration is non-harmful (daily i.p. injections for 7 days lead to peritoneal wall irritation and scaring) and that the same results reported here can be achieved by oral administration of the drug. Also, was the MTF used in this study a long-lasting formulation?
4. Behavioural testing: please, define allodynia and differentiate from mechanical hyperalgesia. Also, there is no justification for starting the tests at day 3 post-surgery, as it is very well established that mechanical hypersensitivity develops very quickly after CCI (within 2 days at the most after surgery). Also, a behavioural baseline should have been obtained prior to any interventions. Details of the experimental set-up and gender of the experimenter should be included, as it is also well established that the pain response varies depending on the sex of the experimenter. Why is there no mention to acclimation? This whole section is far too sketchy.
5. ELISA’s: what is the rationale behind the choice of times for collection of blood samples and sciatic nerves? These times, on top of looking fairly at random, do not coincide with the timing of the behavioural experiments, rendering any possible link between changes in pro-inflammatory markers and behavioural read outs completely impossible.
6. IHC: this section suffers from many important shortcomings. To begin with, perfusion and fixation for 2 days with PFA alone has been shown to render high background levels and is sub-optimal for PNS staining. Second, a gradient sucrose from 10 to 30% should have been used. Third, 20 µm sections are extremely thick for sciatic nerve preparations; 5 to 7 µm are far better suited for quali-quantitative measurements. All the RRID’s are lacking and validation data for all antibodies is missing. Negative controls by means of excluding the primary are barely acceptable and are definitely not proof of antibody specificity. The guidelines of VICTORy should have been followed strictly. As it stands, and given the very poor quality of the images (over-stained, over-saturated, too thick sections) this whole section is unreliable and must be completely re-done.
7. Image analysis: this is completely flawed and unacceptable. A lot more images and from at least 5 animals per group, time and condition should be examined. % of relative fluorescent intensity should be calculated, using background correction and a cut-off percentage for positive and negative staining by means of plotting subjective scores vs. objective measurements. Image J is a blunt tool to do these measurements.
8. WB: I wonder how the authors managed to make a protein extraction of exactly 50 µg of sciatic nerve tissue – to my knowledge, that is nearly impossible. An n of 3 is far too small to get meaningful data and it is an n value that fails to pass any statistical test for predictive power. Method of quantification of relative intensity of WB should be included. Positive and negative controls are missing. Full WBs should be shown, as there is strong background and the quality of the gels is really low.
9. Statistical analyses: include all the assumptions taken into account in the calculations of sample size of each experimental group. State explicitly the results of such calculations. Detail methods of randomisation (if not used, also state that). Include explicit reference to whether testing for outliers was performed. Indicate whether the data sets were tested for normality and how. Indicate what level of confidence was taken as significant.
10. Figure 1A is extremely hard to read and interpret because too many conditions are included in the same time-course, relevant data from baseline should be part of the plot and data from critical days 1 and 2 are missing. The choice of colour is not good. Please, bear in mind that journal guidelines strongly recommend that authors select a color panel that is suited for colour-blind reviewers and readers alike. Some of the differences shown as statistically significant are very small and some were obtained using t-students for as pair of data points. This is completely not valid. I would recommend that the authors plot ipsilateral and contralateral data separately, and that they provide the missing data as well as a better Y-axis to fully appreciate the magnitude of the SEM in each data point. Also, it is standard in the filed that PWT be reported, and in any event, values lower than 2 g are considered allodynia, while values between 6 and 2 correspond to mechanical hyperalgesia. The changes in metabolic parameters have been reported before for the administration of metformin. This would be a valuable contribution if data from the full-time course of treatment are shown. How do the authors explain the changes in glucose levels? The changes in Vit 12 levels are interesting but naïve values for the same time points and more importantly in sham mice (both males and females) at the same time-points should be included. The omission of the experimental groups of sham and sham+MTF for both sexes is a serious shortcoming.
11. Figure 2: images are of very low quality, poor staining, highly unspecific, over-saturated and impossible to clearly distinguish between nodes of Ranvier and the presence/absence of myelinated and, more relevant for neuropathic pain, non-myelinated IB4+ and IB4- fibers (that should have been used as counterstaining instead of other less relevant proteins). The values in the naïve animals fail to coincide with the actual proportion of myelinated fibers in the sciatic nerve. Overall, I find this figure of little use, and the quantification flawed.
12. The long list of cytokines and chemokines and the reported changes is interesting as a descriptive starting point. However, it lacks any meaning in the context of this study as there is no demonstration that these changes are the sole result of the action of MTF. Furthermore, they can be merely part of sex-associated differences in the kinetics of the relevant repair mechanisms, including the elevation of anti-inflammatory mediators like Il-10 that increases in males and females, and in the latter does so with and without MTF. I find that this data set adds nothing of value to this research and it looks very much as an afterthought. Without specific manipulation of the levels of the altered cytokines and a direct demonstration that the observed changes are responsible for the behavioural phenotypes observed, they have very little value.
13. The discussion is long, winded and full of assumptions and not proven associations between players that belong to various processes that are arguably related from the metabolic point of view. But, and most importantly, the reported changes (even if we assume that they are valid despite the flaws in the experimental design and technical execution) in no way prove that any sex-differences observed in the response to MTF after CCI are caused by these alterations.
14. Overall, the lack of proper nociceptive markers, examination of unmyelinated fibers that contribute far more to the responses to noxious stimuli and the disconnection between the immune panel and the rest of the study strike me as serious limitations to this study.
Minor:
1. It is standard in the field to abbreviate neuropathic pain as “NP”. Please stick to that.
2. The introdution is lengthy and have several typos that need to be corrected. The rationale for the study needs to be explained more clearly.
Author Response
R: The study by de Angelis et al. aims at examining the role of metformin as a treatment for neuropathic pain induced by CCI in male and female mice. The authors propose that the glucose-lowering medicine may have different efficacies that can be explained by sex-dependent changes in APMK-driven pathways and autophagy. They also examined levels of various phenotypic markers often associated with peripherally driven repair after nerve injury and also levels of chemokines and cytokines that may be involved in this process.
Albeit the topic of neuropathic pain and the so-called gender gap is timely and relevant, this study has many important shortcomings that dampen my enthusiasm for this work.
My main concerns follow, not necessarily in order of importance:
- Animals: exact number of animals used per group and experiment must be included and justified. Method of blinding (single or double?) must be detailed.
AU: We entirely agree with the reviewer. Indeed, the precise number of animals used for each experiment is already reported in each legend to figure, as well as is reported the full statistical analysis with the corresponding degrees of freedom (DF – derived from number of variables and number of subjects). In any case, to simplify the reading we have now added this information where lacking, in the methods section. With regard to rationale/validation, we have reported (statistic section) that “Concerning the in vivo experiments, we operated in advance the sample size estimation by the implementation of a Power analysis (GPower 3.1)”.
Moreover, we have now included the rationale for the ex-vivo experiments (based on our previous experience and in agreement with animals’ ethic and the 3 “R” principles). As for blinding methods, we previously specified in the “animals” section that “Testing was performed blind as for treatment group to which each subject belonged” (the drug supplier was a person different from the researcher that performed the tests who was blind to the treatment) as well as double blind is considered only for human (blind for experimenters and blind for patients: mice are considered blind per se) .
.
R: 2. Surgery: indicate the drugs rather than the commercial names of the anaesthetic agents used. Include relevant citations for the case of the combination of Tiletamine and Zolazepam having moderate to strong analgesia that lasts far longer than the anaesthesia they induce. This is a potentially contaminating factor when conducting behavioural assessments that are reflex based. Why have the authors omitted a sham operated group as a control?
AU: All the procedures regarding animals use, especially for anaesthesia and surgery are subjected to the Italian (very restrictive and careful to animal’s lab wellness) and European law (DLGs n.26 of 04/03/2014, application of the European Communities Council Directive 2010/63/UE) authorization. Different medical commissions (both intramural and extra-mural (Istituto Superiore di Sanità) authorized and approved anaesthetic drugs, which were considered safe for mice and suitable for the surgery. Moreover, the anaesthetics are under veterinary supervision. Different relevant publications have used such drug combination (i.e. https://www.nature.com/articles/ncomms15292#Sec10). CCI surgical procedure doesn’t last more of 10 minutes for animal.
We have now added in addition to the brand name, also the active substance zoletil (tiletamine and zolazepam, 100 mg ml−1, 0.5 ml kg−1) and rompun (xylazine, 20 mg ml−1, 0.5 ml kg−1)
Regarding the “contamination” problem, we would remind that the behavioural tests start 3 days after surgery and last two months.
Finally, concerning the sham operated mice, we consider that the contralateral hindpaw is a superior control as compared to the hypothesis to operate (and to induce sufferance) an additional number of animals (at least 10 animals) to obtain matching results (“3R principle”). Our long-lasting experience in the study of experimental neuropathic pain in preclinical protocols has suggested this approach to reduce distress and number of animal utilized.
R: 3. Drugs: i.p. injection of metformin is a highly unusual method of delivery, as this drug is essentially administered orally to almost all patients. Also, i.p. injection is painful and often less well tolerated than the oral administration. The latter has the advantage of replicating far more closely the pharmacodynamics and pharmacokinetics of the drug as it is used in the clinical context. Authors show demonstrate that this form of administration is non-harmful (daily i.p. injections for 7 days lead to peritoneal wall irritation and scaring) and that the same results reported here can be achieved by oral administration of the drug. Also, was the MTF used in this study a long-lasting formulation?
AU: Oral administration in mice (gavage) is particularly stressful and harmful, especially if repeated on a daily routine. On the other hand to test drugs and their systemic efficacy, the best choice is the intraperitoneal route of administration. Nevertheless, to circumvent or minimize stress and potential side effects, right and left side are used on alternate days. A vast number of pharmacological studies as well as drug discovery has been and is now produced via the IP route in mice and rats, also for very long chronic regimen of administration. In our experience as well as on the basis of most of the scientific literature no particular drawbacks have never emerged (i.e. Proc Natl Acad Sci U S A. 2007 Feb 20;104(8):2985-90. doi: 10.1073/pnas.0611253104). Regarding MTF, different studies reported IP repeated administration in mice (https://doi.org/10.1016/j.phrs.2018.10.027 https://doi.org/10.1371/journal.pone.0100701)
In any case, we carefully checked possible MTF side effects as evidenced by changes in body weight, Vit B12 levels, glycemia and TGs and the general health status of the mice (e.g., health status of hair coat).
Aim of this work was not the validation/confirmation of MTF effects in mice as compared to patients (and therefore consider pharmacokinetic and pharmacodynamics, which is now an established knowledge) but to study and possibly disclose sex differences in the neuropathic pain and discrepancy in response to the drug in mice.
R: 4. Behavioural testing: please, define allodynia and differentiate from mechanical hyperalgesia. Also, there is no justification for starting the tests at day 3 post-surgery, as it is very well established that mechanical hypersensitivity develops very quickly after CCI (within 2 days at the most after surgery). Also, a behavioural baseline should have been obtained prior to any interventions. Details of the experimental set-up and gender of the experimenter should be included, as it is also well established that the pain response varies depending on the sex of the experimenter. Why is there no mention to acclimation? This whole section is far too sketchy.
AU: Allodynia signifies a painful response to a non-painful stimulus, while hyperalgesia is an exacerbated painful response to a painful stimulus. To clarify this point, we insert the allodynia definition in methods section.
We started to test 3 days after surgery because, in accordance with the guideline of wellness of the animals lab, 3 days are the least period allowing mice to full restore from the surgery. This option doesn’t affect the validity of results since neuropathy in mice is prolonged at least for 60 days in male subjects and over in females (Vacca, V.; et al. 17Beta-Estradiol Counteracts Neuropathic Pain: A Behavioural, Immunohistochemical, and Proteomic Investigation on Sex-Related Differences in Mice. Sci. Rep. 2016, 6, doi:10.1038/srep18980; Vacca, V, et al Higher Pain Perception and Lack of Recovery from Neuropathic Pain in Females: A Behavioural, Immunohistochemical, and Proteomic Investigation on Sex-Related Differences in Mice. Pain 2014, 155, doi:10.1016/j.pain.2013.10.027). The pharmacological treatment started immediately after CCI, and different measurements such as in TNFa have been analysed at very early time-points (i.e. 6h and 24h). Baseline is present in Fig. 1A, left graph
As demonstrated by Mogil et al the experimenter gender influences behavioural response and although this can happen, male experimenters cannot be excluded from the research. Nevertheless, we believe that is not correct to insert the sex of experimenters in the methods, since some experimenters could be identified in different ways.
Thank you for pointing out the lacking of the acclimation method, which we have now added.
R: 5. ELISA’s: what is the rationale behind the choice of times for collection of blood samples and sciatic nerves? These times, on top of looking fairly at random, do not coincide with the timing of the behavioural experiments, rendering any possible link between changes in pro-inflammatory markers and behavioural read outs completely impossible.
AU: ELISA has been utilized to evaluate the following agents: TNFα, VIT B12 and SIRT1, and the time-points considered are: baseline (tissue derived from naïve animals), 6h, 24h, D3, D7.
This time-points of analysis allow us to evaluate fluctuation of the molecules taken into consideration as response to: neuropathy, treatment and sex, in combination or alone.
We are aware that some of these (i.e. TNFa) can change very quickly after inflammation (Neuroscience Letters 436 (2008) 210–213) and others have never investigated before (neither in nerves nor in blood). Thus, we started the treatment immediately after ligature, which is important to understand if MTF interferes with the factors investigated. Moreover we ended the treatment 7 days after ligature (during the first seven days Schwann cells autophagy reaches the peak), and therefore this time-point allows to establish the final modulation of these factors at the end of the treatment period.
These experiments, although helping to understand the changes in the behavioural response to neuropathy, have also been developed (in combination with additional factors) to understand the molecular mechanisms through which MTF acts (excepted for VIT B12 to monitor possible side-effect), and generates the sex differences reported and discussed in the paper.
R: 6. IHC: this section suffers from many important shortcomings. To begin with, perfusion and fixation for 2 days with PFA alone has been shown to render high background levels and is sub-optimal for PNS staining. Second, a gradient sucrose from 10 to 30% should have been used. Third, 20 µm sections are extremely thick for sciatic nerve preparations; 5 to 7 µm are far better suited for quali-quantitative measurements. All the RRID’s are lacking and validation data for all antibodies is missing. Negative controls by means of excluding the primary are barely acceptable and are definitely not proof of antibody specificity. The guidelines of VICTORy should have been followed strictly. As it stands, and given the very poor quality of the images (over-stained, over-saturated, too thick sections) this whole section is unreliable and must be completely re-done.
AU: The IHC protocol is used in our lab since 2010 and has been carefully tested and standardized, allowing to obtain high-quality images published in more than 50 peer-reviewed publications and gaining also a cover from an important journal. The thickness of the section not affect measurement, since we use confocal microscopy to acquire images and therefore we can select the plan of acquisition. Concerning the specificity of antibodies, they are tested at the beginning (the first time that we buy them or a new batch) and for the most part they are antibodies continuously utilized in our lab (as demonstrated by many publications). They are all commercial antibodies for which all information is inserted by the vendors or developers in the RRID, and available for anyone interested. Generally is unnecessary to insert in the paper the RRID since all commercial information for each antibody are available. (incidentally, this is a practise followed by any paper). VICTORy guidelines suggest: (1) the validation of antibodies, (2) their identification, (3) communication and controls, (4) the training of potential users, (5) the transparency of original equipment manufacturer (OEM) marketing agreements, and (5) in a more widespread use of recombinant antibodies (together denoted the ‘VICTOR’ approach). Indeed, we entirely adhere to these guidelines?
R: 7. Image analysis: this is completely flawed and unacceptable. A lot more images and from at least 5 animals per group, time and condition should be examined. % of relative fluorescent intensity should be calculated, using background correction and a cut-off percentage for positive and negative staining by means of plotting subjective scores vs. objective measurements. Image J is a blunt tool to do these measurements.
AU: As the reviewer can verify, and as required from the editorial format of the journal, images are inserted inside the text (small format and low quality). Later, high quality images will be uploaded individually after revision. To analyse the images two different approaches have been followed: cell count (such as in the case of macrophages and nuclei) in which background and saturation are not essential, and in the second case we used the RGB method that considers the pixels and in this case this method requires to subtract the background (we have now better described this point in the method section). Image J is not “a blunt tool to do these measurements” and is widely used and accepted. Perhaps, someone should ask this to NIH developers and maybe write a methodological paper on this matter.
R: 8. WB: I wonder how the authors managed to make a protein extraction of exactly 50 µg of sciatic nerve tissue – to my knowledge, that is nearly impossible. An n of 3 is far too small to get meaningful data and it is an n value that fails to pass any statistical test for predictive power. Method of quantification of relative intensity of WB should be included. Positive and negative controls are missing. Full WBs should be shown, as there is strong background and the quality of the gels is really low.
AU: We are sorry for the misunderstanding, probably we hadn’t in depth detailed the WB method. To explain: We have pooled 3 nerves deriving from 3 different animals for each single experiment and we have exactly pulled 50 ug of total protein lysate for each gel. Now, we better specify in the text.
Regarding method of quantification it is already present in the text: “….Luminescent bands were imaged with autoradiography (X-ray) films (UltraCruz; Santa Cruz Biotechnology) and then scanning into a digital format. The β-actin bands intensity were used as a control for equal protein loading and measured for densitometric analysis using ImageJ 1.49r software (Wayne Rasband, National Institutes of Health)”. We can specify in the text that the ratio value is obtained by ampk/actin.
Regarding the N we utilize with rigour the rule of minimum number of animals necessary (in accordance we 3R) to have a good data. We would argue that we have 6 groups of animals (naïve M and F; Vehicle M and F; MTF M and F), we proposed a great number of ex-vivo experiments (IHC, ELISA, WB, antibody array) which require different methods of collection and conservation, we have different time-points (BL, 6h, 24h, D3, D7) and that for each time point at least 3 animals have been sacrificed, this imply that at least 360 animals have been utilized. Therefore we increase the N per group when we observed a strong variability in the data and we repeat the experiments. Also with small animals groups is possible to have statistical significance and “predictive power” since the appropriate tests have been utilized as we reported in the manuscript “…while for small samples (N<5 animals) and groups >3, non-parametric analysis was performed by Kruskall-Wallis”.
Starting from the assumption that AMPK and p-AMPK are physiologically expressed in all tissues, we can considered negative control (very few presence) for p-AMPK the naïve animals while as positive (great amount/sure activation) animals with caloric restriction (able to active AMPK) as demonstrated in previous paper (Coccurello, R et al Effects of Caloric Restriction on Neuropathic Pain, Peripheral Nerve Degeneration and Inflammation in Normometabolic and Autophagy Defective Prediabetic Ambra1 Mice. PLoS One 2018, 13, doi:10.1371/journal.pone.0208596.)
R: 9. Statistical analyses: include all the assumptions taken into account in the calculations of sample size of each experimental group. State explicitly the results of such calculations. Detail methods of randomisation (if not used, also state that). Include explicit reference to whether testing for outliers was performed. Indicate whether the data sets were tested for normality and how. Indicate what level of confidence was taken as significant.
AU: The whole experimental design, thus including the sample size required, has been developed before any “in vivo” laboratory experiment. Hence, we performed an “a priori” power analysis whose results were the following:
F tests - ANOVA: Repeated measures, within-between interaction
Analysis: A priori: Compute required sample size
Input: Effect size f = 0.2000000
α err prob = 0.05
Power (1-β err prob) = 0.95
Number of groups = 4
Number of measurements = 15
Corr among rep measures = 0.5
Nonsphericity correction ε = 1
Output: Noncentrality parameter λ = 43.2000000
Critical F = 1.4117114
Numerator df = 42.0000000
Denominator df = 448
Total sample size = 36
Actual power = 0.9518883
As showed, to achieve a statistical power of 0.95, and keep the α err within 0.05, the result of analysis reports about 36 subjects that for the 4 experimental conditions we used means N=9 subjects x group (36/4=9). We did more by using 10 subjects per group. In any case, such power analysis cannot be inserted in the txt of the manuscript; indeed there is no publication reporting that.
The level of confidence taken as significant is indicated (α=0.05), while testing for outliers has been not performed in particular because within this context and methodology there is no possibility to test that. Homoscedasticity is an assumption implicit to the use of ANOVA analysis. These data are normally distributed and there is no reason to assume the non-normality; every single research paper investigating experimentally-induced allodynia performed on these data ANOVA analysis.
R: 10. Figure 1A is extremely hard to read and interpret because too many conditions are included in the same time-course, relevant data from baseline should be part of the plot and data from critical days 1 and 2 are missing. The choice of colour is not good. Please, bear in mind that journal guidelines strongly recommend that authors select a color panel that is suited for colour-blind reviewers and readers alike. Some of the differences shown as statistically significant are very small and some were obtained using t-students for as pair of data points. This is completely not valid. I would recommend that the authors plot ipsilateral and contralateral data separately, and that they provide the missing data as well as a better Y-axis to fully appreciate the magnitude of the SEM in each data point. Also, it is standard in the filed that PWT be reported, and in any event, values lower than 2 g are considered allodynia, while values between 6 and 2 correspond to mechanical hyperalgesia. The changes in metabolic parameters have been reported before for the administration of metformin. This would be a valuable contribution if data from the full-time course of treatment are shown. How do the authors explain the changes in glucose levels? The changes in Vit 12 levels are interesting but naïve values for the same time points and more importantly in sham mice (both males and females) at the same time-points should be included. The omission of the experimental groups of sham and sham+MTF for both sexes is a serious shortcoming.
AU: We previously replied to this query (a better visualization will be available when images will be uploaded in high-quality and dimension). BL cannot be included in the same graph because is split in two groups: females and males. Experimental groups are randomly assigned after surgery. All conditions are required to be inserted in the same graph because of the multiple comparisons and interactions analysed. Day 1 and 2 are not critical days as above explained.
R: The choice of colour is not good. Please, bear in mind that journal guidelines strongly recommend that authors select a colour panel that is suited for colour-blind reviewers and readers alike.
AU: Pink and light-blue are not mismatched by “colour-blind reviewers and readers”. They could simply perceived different colours (depending of the kind of colour-blind) but not confuse them. In fact, considering the different kind of deficits: Protan Colour Blindness red-green colour blindness; Deutan Color Blindness misperceptions between colours such as green and yellow, or blue and purple; Tritan Color Blindness characterized by a reduced sensitivity in the blue-sensitive (confuse blue-yellow); Monochromacy and Achromatopsia “no colour” vision
R: Some of the differences shown as statistically significant are very small and some were obtained using t-students for as pair of data points. This is completely not valid. I would recommend that the authors plot ipsilateral and contralateral data separately, and that they provide the missing data as well as a better Y-axis to fully appreciate the magnitude of the SEM in each data point.
AU: As a matter of fact, “differences” are “differences” as long as they have been obtained by a correct statistical analysis; which is, indeed, the case. We do not elect the magnitude of statistical differences. The t student was applied after ANOVA significance as in fig.1. Moreover, plotting separately contralateral and ipsilateral data won’t help or add information. Indeed, just because contralateral paw is used (not by us but in all papers implementing the sciatic nerve ligature as model of NeP) as inner control within the same subject, all the lines are collapsed in the upper part of the figure. What would be the advantage for the reader to see a panel in which only almost identical responses are depicted as control (depicting the response from contralateral hindpaw)?
R: Also, it is standard in the filed that PWT be reported, and in any event, values lower than 2 g are considered allodynia, while values between 6 and 2 correspond to mechanical hyperalgesia.
AU: We found this categorisation reductive and not appropriate in mice. There is a large number of technical reasons for that, often mere technicalities. For instance, at difference with von frey filaments, we operate with a dynamic aesthesiometer which is composed by a single metallic filament (not painful per se) which exerts a time-increasing force (measured in grams) on the plantar surface. The mechanical threshold tends to increase during the time reaching baseline values when neuropathy is restored.
R: The changes in metabolic parameters have been reported before for the administration of metformin. This would be a valuable contribution if data from the full-time course of treatment are shown. How do the authors explain the changes in glucose levels?
AU: Metabolic parameters were measured before (BL) and after the end of MTF treatment (latest administration) (D7), revealing effects of the chronic treatment and potential side-effects. Although the metabolic measures are minimally invasive (few blood drops from the tail), repeated manipulation are stressful. We assume the conduct inspired by a minimal and unavoidable stress interference in behavioural test. We have clearly explained the changes in glucose levels in our previous papers (Coccurello, R et al Effects of Caloric Restriction on Neuropathic Pain, Peripheral Nerve Degeneration and Inflammation in Normometabolic and Autophagy Defective Prediabetic Ambra1 Mice. PLoS One 2018, 13, doi:10.1371/journal.pone.0208596.), as described in the present work.
R: The changes in Vit 12 levels are interesting but naïve values for the same time points and more importantly in sham mice (both males and females) at the same time-points should be included. The omission of the experimental groups of sham and sham+MTF for both sexes is a serious shortcoming.
AU: We have previously discussed this aspect about sham animals.
R: 11. Figure 2: images are of very low quality, poor staining, highly unspecific, over-saturated and impossible to clearly distinguish between nodes of Ranvier and the presence/absence of myelinated and, more relevant for neuropathic pain, non-myelinated IB4+ and IB4- fibers (that should have been used as counterstaining instead of other less relevant proteins). The values in the naïve animals fail to coincide with the actual proportion of myelinated fibers in the sciatic nerve. Overall, I find this figure of little use, and the quantification flawed.
AU: We have now uploaded in the journal system high-quality and bigger dimension images allowing to appreciate every details, in particular in the zoomed square. Some areas over-saturated are standard in lesioned areas (up, down and intra-ligatures spaces), and when neurodegeneration is in progress. Because of myelin aggregates, some proteins overproduction, degenerated axons, etc are present and up-regulated. This determines an enhancement of the fluorescence intensity. In fact, this doesn’t occur in staining from naïve sciatic nerve (CTRL images, now added in supplementary material).
IB4+ fibers together with CGRP mark C-fibers (small diameter unmyelinated fibres) which transport nociceptive stimuli (especially thermal) to spinal cord but that in neuropathic pain, allodynia or mechanical hypersensitivity is mediated by Alpha and Beta fibers (myelinated). In any case, the scope of this data were to investigate the effects of MTF as pro-autophagic drug, also on Wallerian degeneration and therefore on demyelination phenomena.
The assertion “The values in the naïve animals fail to coincide with the actual proportion of myelinated fibers in the sciatic nerve” does not consider that values are referring to brightness (calculated on the pixels), not representing a number or a percentage of fibers. Anyhow, the evaluation of brightness is not considered a quantitative method but an evaluation of the marker expression and together with a morphological discussion can be useful in a qualitative evaluation. By figures and their analysis, we can understand and therefore discuss about a greater or lesser nerve degeneration. This is the actual purpose of this experiment.
R: 12. The long list of cytokines and chemokines and the reported changes is interesting as a descriptive starting point. However, it lacks any meaning in the context of this study as there is no demonstration that these changes are the sole result of the action of MTF. Furthermore, they can be merely part of sex-associated differences in the kinetics of the relevant repair mechanisms, including the elevation of anti-inflammatory mediators like Il-10 that increases in males and females, and in the latter does so with and without MTF. I find that this data set adds nothing of value to this research and it looks very much as an afterthought. Without specific manipulation of the levels of the altered cytokines and a direct demonstration that the observed changes are responsible for the behavioural phenotypes observed, they have very little value.
AU: Cytokines and chemokines analysis has been performed in sciatic nerve in the following conditions and separately in males and females: 1) Naïve (representing the BL); 2) Neuropathy; 3) Neuropathy + MTF; and each condition compared with another. Therefore, if there are differences between neuropathy alone and neuropathy + MTF, the effect of variation can be attributable to MTF. Sex-dependent modulation of inflammatory agents is evident by observing the different factors modulated in response to neuropathy. This set of data provided a valuable picture of MTF treatment on inflammatory agents, confirming the indication that MTF increase chemoattractant factors and therefore macrophages activity, also supported by other sets of experiments (TNFa, macrophages, IPA analysis…). The purpose of these experiments were to understand sex-differences associated to innate immunity response to neuropathy after MTF treatment that could be responsible of the behavioural phenotype, as we have discussed in the paper.
R: 13. The discussion is long, winded and full of assumptions and not proven associations between players that belong to various processes that are arguably related from the metabolic point of view. But, and most importantly, the reported changes (even if we assume that they are valid despite the flaws in the experimental design and technical execution) in no way prove that any sex-differences observed in the response to MTF after CCI are caused by these alterations.
- Overall, the lack of proper nociceptive markers, examination of unmyelinated fibers that contribute far more to the responses to noxious stimuli and the disconnection between the immune panel and the rest of the study strike me as serious limitations to this study.
AU: We have very respectfully and, above all, in workwise and in a qualified manner replied to the potential “misunderstandings” and personal opinions expressed in this report.
Minor:
- It is standard in the field to abbreviate neuropathic pain as “NP”. Please stick to that.
This is not always true; usually NP is utilized for “neuropathy” while for example in papers published in the Pain (IASP journal) neuropathic pain is abbreviated as NeP as well as in several clinical reviews (https://journals.lww.com/pain/pages/results.aspx?txtKeywords=NeP; Rev Pain. 2011 Jun; 5(2): 1–2; https://www.tandfonline.com/doi/full/10.1080/03007995.2017.1321532 )
- The introduction is lengthy and have several typos that need to be corrected. The rationale for the study needs to be explained more clearly.
We revised the paper for typos. The “lengthy” introduction was just to properly explain the rationale.
Round 2
Reviewer 2 Report
The authors have resourced mainly to their previous publications to justify their technical choices.
They have not answered satisfactorily either point 5 nor point 6.
I also find disrespectul to be patrosnising - I have worked in the field of pain and with DRG neurons for over 30 yeras and have an excellent understanding of the topic. The authors "mini-lectures" in their responses are uncalled for and unnnecessary.
The overall quality of thw work remains poor. The many important flaws have gone largely ignored, and excuses made for obvious omissions. Still, I understand that increasing the number of mice without excellent reason is not ideal, and respect that contralateral is a better control than sham in this instance.